# Wonder Wins Ways: Curiosity-Driven Exploration through Multi-Agent Contextual Calibration

**Yiyuan Pan    Zhe Liu*    Hesheng Wang***
Shanghai Jiao Tong University
{pyy030406, liuzhesjtu, wanghesheng}@sjtu.edu.cn

## Abstract

Autonomous exploration in complex multi-agent reinforcement learning (MARL) with sparse rewards critically depends on providing agents with effective intrinsic motivation. While artificial curiosity offers a powerful self-supervised signal, it often confuses environmental stochasticity with meaningful novelty. Moreover, existing curiosity mechanisms exhibit a uniform novelty bias, treating all unexpected observations equally. However, peer behavior novelty, which encode latent task dynamics, are often overlooked, resulting in suboptimal exploration in decentralized, communication-free MARL settings. To this end, inspired by how human children adaptively calibrate their own exploratory behaviors via observing peers, we propose a novel approach to enhance multi-agent exploration. We introduce CERMIC, a principled framework that empowers agents to robustly filter noisy surprise signals and guide exploration by dynamically calibrating their intrinsic curiosity with inferred multi-agent context. Additionally, CERMIC generates theoretically-grounded intrinsic rewards, encouraging agents to explore state transitions with high information gain. We evaluate CERMIC on benchmark suites including VMAS, Meltingpot, and SMACv2. Empirical results demonstrate that exploration with CERMIC significantly outperforms SoTA algorithms in sparse-reward environments.

**Code:** https://github.com/PyyWill/CERMIC

## 1 Introduction

Achieving effective exploration in complex Multi-Agent Reinforcement Learning (MARL) settings, particularly those characterized by sparse rewards and partial observability, remains a formidable scientific challenge [7, 43]. Intrinsic motivation, instantiated as artificial curiosity, has emerged as a key ingredient for unlocking autonomous learning by providing self-supervised signals in the absence of immediate extrinsic feedback [29, 38]. This internal drive enables agents to acquire skills and knowledge that support robust, adaptive intelligence.

However, such novelty-seeking algorithms is susceptible to the stochastic environment dynamics or other unlearnable noises (the "*Noisy-TV*" problem) [23]. Existing algorithms mitigate this challenge through uncertainty quantification or by exploiting global information in multi-agent systems. However, these strategies prove insufficient for intelligent agents, particularly heterogeneous ones or those in large-scale systems: *Firstly*, such agents frequently encounter severe partial observability, rendering inaccurate uncertainty estimates due to insufficient replay experiences [20]; *Secondly*, in decentralized execution without effective communication, agents struggle to form accurate beliefs

---

*Corresponding authors. The authors are with the School of Automation and Intelligent Sensing, Shanghai Jiao Tong University, the Key Laboratory of System Control and Information Processing, Ministry of Education of China, Shanghai 200240. Zhe Liu is also with the National Key Laboratory of Human-Machine Hybrid Augmented Intelligence, Institute of Artificial Intelligence and Robotics, Xi'an Jiaotong University.

39th Conference on Neural Information Processing Systems (NeurIPS 2025).

about others' latent states, undermining methods that presuppose shared inter-agent information. [13]. Altogether, these limitations highlight a critical need for exploration mechanisms that are robust to partial observable and communication-less environments.

Insights from human cognitive development suggest a pathway forward: children rapidly adapt to new social games not only through solo trial-and-error, but also by observing peers, inferring intentions, and selectively imitating successful strategies [42, 16]. Such form of social learning, often driven by an innate curiosity about 'why' others act as they do, allows for swift coordination and an understanding of task dynamics, even without complete information or explicit instruction [21, 30]. Naturally, the success of this human-centric learning process motivates translating its core principles to Multi-Agent Systems (MAS). Therefore, this paper seeks to answer:

> *How can robust, context-aware calibration of intrinsic curiosity unlock truly autonomous and effective exploration for MARL agents in challenging real-world settings?*

To this end, we propose **C**uriosity **E**nhancement via **R**obust **M**ulti-Agent **I**ntention **C**alibration (CERMIC), a modular, plug-and-play component designed to augment existing MARL exploration algorithms. Based on the Information Bottleneck (IB) principle [41, 40], CERMIC learns a multi-agent contextualized exploratory representation that steers exploration toward semantically meaningful novelty, and filters unpredictable and spurious novelty. Specifically, it incorporates a graph-based module to model the inferred intentions of surrounding agents and use the context to calibrate raw individual curiosity signal at a given coverage level. At each episode, CERMIC yields a loss for self-training and a theoretically-grounded intrinsic reward for exploration. We empirically validate CERMIC by integrating it with various MARL algorithms and evaluating its performance across challenging benchmark suites. In summary, our contributions are threefold:

- We introduce CERMIC, a novel framework that empowers MARL agents with socially contextualized curiosity. Inspired by developmental psychology, this offers a novel perspective on the crucial challenges of effective exploration in sparse-reward settings.

- We propose a robust and controllable multi-agent calibration mechanism in challenging partially observable and communication-limited environments. CERMIC allows for adaptive tuning based on the learned reliability of the intention graph, effectively dampening exploration instability often plaguing vanilla novelty-seeking agents.

- We deliver CERMIC as a lightweight, readily integrable module and demonstrate consistent gains over strong baselines across standard benchmarks under sparse rewards.

## 2 Related Work

**Naïve Exploration in RL.** Early exploration strategies in reinforcement learning (RL), such as $\epsilon$-greedy or Boltzmann exploration, gained popularity due to their simplicity and ease of integration into various algorithms [25, 35, 10]. However, these "naive" exploration methods often perform suboptimally, particularly in challenging scenarios characterized by sparse rewards or deceptive local optima. Effectively, the agent explores by taking largely random sequences of actions, which, especially in continuous state and action spaces, makes comprehensive coverage exceptionally difficult. Even in the foundational setting of multi-armed bandits (MAB) in continuous spaces [6], more theoretically sound yet still model-free exploration strategies like Thompson Sampling (TS), Upper-Confidence Bound methods, and Information-Directed Sampling (IDS) have been developed [37, 33, 11]. However, while these methods offer improvements over purely random approaches, their efficacy remains limited when progress requires discovering semantically meaningful novelty far from the agent's initial experience distribution. This motivates the development of intrinsic reward that can provide more targeted and adaptive exploration signals.

**Curiosity-Driven Exploration** Intrinsic rewards are a powerful tool for driving exploration in RL, especially in sparse-reward single-agent tasks [29, 4]. Early methods often quantified novelty via visitation metrics like pseudo-counts or hashing. Subsequent approaches broadened this by measuring surprise through prediction errors (of transitions or features), state marginal matching, uncertainty estimates, or even TD errors from random reward predictors [18, 32]. Then, multi-agent exploration, however, presents unique inter-agent dynamics requiring tailored intrinsic rewards. Some works have explored inter-agent observational novelty or influence over others' transitions/values

[44]. Others have investigated shared intrinsic signals (e.g., summed local Q-errors) or information-theoretic objectives like maximizing trajectory-latent mutual information for behavioral diversity[27]. Techniques such as goal-based methods with dimension or observation-space goal selection also aim to manage state space complexity [12, 22]. However, many multi-agent intrinsic rewards implicitly rely on centralized training signals, privileged communication, or access to global state—assumptions that break down under decentralized execution. In addition, severe partial observability and heterogeneous policies can mis-calibrate uncertainty-based novelty estimates, while curiosity mechanisms that treat all unexpected events uniformly tend to amplify spurious novelty and overlook socially informative cues encoded in peers' behaviors. To address these shortcomings, we propose CERMIC, a novel framework for robustly calibrating intrinsic curiosity via learned models of inferred inter-agent intentions.

# 3 Background

**Problem Setup.** We consider a MAS operating within a communication-less, partially observable Markov Decision Process (POMDP) [5]. Formally, the environment is described by a tuple $(\mathcal{O}, \mathcal{A}, \mathbb{P}, \mathcal{R}, N, \gamma)$. At each timestep t, each agent $i \in N$ receives a local observation $o_t^i \in \mathcal{O}$ and select an action $a_t^i \in \mathcal{A}$ via decentralized policies, which induces a state transition. All the agents receive a shared extrinsic rewards $r_t^e \in \mathcal{R}$ after executing the joint action. In this paper, we focus on the individual agent and omit index $i$ throughout; We use uppercase letters to denote random variables and lowercase letters to denote their realizations.

**Preliminary.** The Information Bottleneck (IB) principle [2, 17] guides the learning of a compressed representation $Z$ of an input $X$ that maximally preserves information about a target $Y$. This is achieved by optimizing the trade-off:

$$\max I(Z; Y) - \alpha I(X; Z)$$

where $I(\cdot; \cdot)$ denotes mutual information. The first term, $I(Z; Y)$, measures how much information the representation $Z$ contains about the target variable $Y$, ensuring that $Z$ preserves the relevant predictive information for $Y$; The second term, $I(X; Z)$, quantifies the amount of information that $Z$ retains about the original input $X$, compelling $Z$ to be a succinct summary of $X$. In our work, the IB principle serves as a foundational concept.

**Method Overview.** CERMIC processes transition experiences $(s_t, a_t, s_{t+1}, r_{t-1}^e)$ to generate intrinsic rewards and guide exploration, where state embeddings $s_t$ and $s_{t+1}$ are obtained from observations $o_t$ and $o_{t+1}$ using encoders with the same structures. To ensure stable representation learning, the parameters of $o_{t+1}$'s encoder ($\theta^m$) is updated via a momentum moving of $o_t$'s ones ($\theta$). The core of CERMIC is to learn a latent representation $x_t$, parameterized as a Gaussian distribution conditioned on the current state-action pair: $x_t \sim g_\phi(s_t, a_t)$. Following the IB principle, CERMIC seeks to maximize mutual information $I(X_t; S_{t+1})$ to retain predictive information of novel states and encourage *exploration*, while minimizing $I(X_t; [S_t, A_t])$ to *exploit* information about the current context and gain a compressed representation. The compression process $\min I(X_t; [S_t, A_t])$ incorporates robust calibration mechanisms leveraging multi-agent context. The overall objective is formulated as Eq. (1):

$$\max I(X_t; S_{t+1}) - \alpha I(X_t; [S_t, A_t]) \tag{1}$$

where $\alpha$ is a Lagrange multiplier. In what follows, this work tries to address two key questions: (i) *How is* CERMIC *trained (Section 4.3)?* (ii) *How does* CERMIC *generate intrinsic rewards to drive exploration (Section 4.4)?*

# 4 Method

## 4.1 Novelty-Driven Exploration

To drive exploration, we aim to maximize the mutual information $I(S_{t+1}; X_t)$. However, direct optimization is intractable. Instead, we resort to maximizing a tractable variational lower bound.

$$
\begin{aligned}
I(X_t; S_{t+1}) &= \mathbb{E}_{p(x_t, s_{t+1})} \left[ \log \frac{p_\phi(s_{t+1}|x_t)}{p(s_{t+1})} \right] + \mathcal{D}_{\text{KL}}[p(s_{t+1}|x_t) \| p_\phi(s_{t+1}|x_t)] \\
&\geq \mathbb{E}_{p(x_t, s_{t+1})}[\log p_\phi(s_{t+1}|x_t)] + \mathcal{H}(S_{t+1})
\end{aligned}
\tag{2}
$$

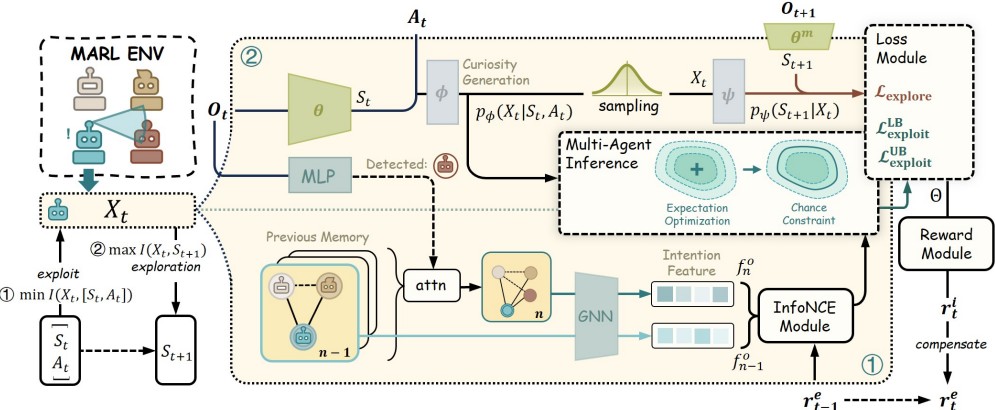

Figure 1: **Workflow of CERMIC**. (**Left**) Illustrates CERMIC's per-timestep objective. (**Right**) Depicts the network architecture. By projecting the raw observation $o_t$ into a lower-dimensional embedding $s_t$, subsequent computations within CERMIC operate on compact representations, contributing to its overall computational efficiency.

where $p_\phi(s_{t+1}|x_t)$ is a variational encoder parameterized by $\phi$ to approximate the unknown true conditional distribution $p(s_{t+1}|x_t)$. Then, by the non-negativity of the KL-divergence, we obtain the second line expression. Since the entropy of the true next state distribution $\mathcal{H}(S_{t+1})$ is independent of the model parameters $\phi$, it's equivalent to discard this term. Thus, the exploration objective simplifies to maximizing a log-likelihood loss under the variational approximation.

$$\mathcal{L}_{\text{explore}} \triangleq \mathbb{E}_{p(x_t, s_{t+1})}[\log p_\phi(s_{t+1}|x_t)] \tag{3}$$

## 4.2 Multi-Agent Contextualized Exploitation

Similarly, to minimize the intractable mutual information $I(X_t; [S_t, A_t])$, we optimize its variational upper bound. Following prior work [2], we introduce a Gaussian variational approximation $q(x_t) \sim N(0, \mathbf{I})$ for the intractable marginal distribution $p(x_t) = \int p(x_t|s_t, a_t)p(s_t, a_t)ds_t da_t$:

$$I([S_t, A_t]; x_t) = \mathbb{E}_{p(s_t, a_t, x_t)} \log \frac{p(x_t|s_t, a_t)}{q(x_t)} - \mathcal{D}_{KL}[p(x_t)\|q(x_t)] \leq \mathbb{E}_{p(s_t, a_t, x_t)} \Psi_t. \tag{4}$$

where the inequality holds due to the non-negativity of KL-divergence. For conciseness, let $\Psi_t = \log p(x_t|s_t, a_t)/q(x_t)$. The mean $\mu_\Psi$ and variance $\Sigma_\Psi$ of $\Psi_t \sim \mathcal{P}^\Psi(\mu_\Psi, \Sigma_\Psi)$ can be numerically computed through $p(x_t|s_t, a_t)$ and $q(x_t)$. However, directly computing the expectation in Eq. (4) remains challenging due to the distribution shift caused by replacing $p(x_t)$ and the high-dimensional Monte-Carlo (MC) sampling required [36]. To address this, we enforce an upper bound $\bar{c}$ on $\Psi_t$ to relax the minimization term using a chance constraint:

$$\mathbb{P}_r(\Psi_t \leq \bar{c}) \geq 1 - \epsilon, \ \mathbb{P} \in \mathcal{P}^\Psi. \tag{5}$$

Crucially, to imbue this compression process with awareness of other agents in our communication-less MAS setting, we condition the chance constraint on an inferred multi-agent contextual feature, $f_n^o$. This feature $f_n^o$ is encoded from an underlying intention modeling of other agents and the specific architecture of this intention modeling can be varied. In our primary exposition, we exemplify the intention model as a dynamic graph, $G_n$ (additional analyses on the effects of pre-trained intention models and other forms of intention modeling can be found in Section 5.3 and Appendix G). The nodes in $G_n$ encapsulate predictive state representations for each agent, while the edges represent their relative spatial relationships. We maintain a memory queue of the intention modeling over time, and the graph $G_n$ will be advanced to $G_{n+1}$ only when other agents are detected by a simple MLP module, ensuring efficient computation under partial observability (the trained MLP can identify and match different agents). The generation of $G_n$ at each step utilizes an attention mechanism over historical intention modeling and current observations. Subsequently, a Graph Neural Network

(GNN) [34] processes $G_n$ to yield the contextual feature $f_n^o$, as detailed in Appendix A. Ultimately, regardless of its precise origin, this inferred context $f_n^o$ is integrated into the chance constraint via distributional robustness:

$$\inf_{\mathbb{P}_r \in \mathcal{P}_{\text{cal}}^{\Psi}} \mathbb{P}_r(\Psi_t \leq \bar{c}) \geq 1 - \epsilon \tag{6}$$

where

$$\mathcal{P}_{\text{cal}}^{\Psi} = \left\{ \mathbb{P}_r \in \mathcal{P}(\mathbb{R}) : \begin{array}{c} (\mathbb{E}_{\mathbb{P}_r}[\Psi_t] - \mu_\Psi(f_n^o))^2 \Sigma_\Psi^{-1} \leq \gamma_1 \\ \mathbb{E}_{\mathbb{P}_r}[(\Psi_t - \mu_\Psi)^2] \leq \gamma_2 \Sigma_\Psi \end{array} \right\}.$$

Here, $\gamma_1 > 0$ and $\gamma_2 > \max\{\gamma_1, 1\}$ are constants, and the mean value now is dependent on the feature $f_n^o$. The first inequality of $\mathcal{P}_{\text{cal}}^{\Psi}$ represents the true mean value is within an ellipsoid centered at $\mu_\Psi(f_n^o)$ with a bound $\gamma_1$, while the second one constrains the true variance with a bound $\gamma_2$. The robustness of the set has been proved using McDiarmid's inequality [8], and the optimal $\gamma_1, \gamma_2$ can be correspondingly derived. Next, this robust chance-constraint formulation can then be tractably converted into the following second-order cone loss in Eq. (7) via Cantelli's inequality [26] (see Appendix B for proof), resulting in a loss function where $\beta$ is a constant regarding $\gamma_1, \gamma_2, \epsilon$. In essence, minimizing this loss promotes exploitation within the defined robust calibration set.

$$\mathcal{L}_{\text{exploit}}^{\text{UB}} \triangleq \text{ReLU}(\mu_\Psi(f_n^o) + \beta\sqrt{\Sigma_\Psi} - \bar{c}). \tag{7}$$

To ensure the robustness and stability of the curiosity system in the worst case, a lower bound loss is also applied as follows, transformed from $\mathbb{P}_r(\Psi_t \geq \underline{c}) \geq 1 - \epsilon$. We can set task-agnostic $\bar{c}$ and $\underline{c}$ after normalizing $\Psi_t$.

$$\mathcal{L}_{\text{exploit}}^{\text{LB}} \triangleq \text{ReLU}(-\mu_\Psi(f_n^o) + \beta\sqrt{\Sigma_\Psi} - \underline{c}). \tag{8}$$

**Task-Adaptive Calibration.** This paragraph details the implementation of the calibration mechanism for the mean parameter $\mu_\Psi(f_n^o)$. We formulate the calibrated mean as a network-parameterized function $h$, regarding the inferred multi-agent context $f_n^o$, a task-adaptive factor $\gamma$, and the original mean $\mu_\Psi$. $\gamma$ is the mutual information between the inferred intention $f_n^o$ and the preceding context composed of the previous intention $f_{n-1}^o$ and the received extrinsic reward $r_{t-1}^e$.

$$\mu_\Psi(f_n^o | \gamma = I([r_{t-1}^e, f_{n-1}^o]; f_n^o)) = h(\gamma f_n^o, \mu_\Psi) \tag{9}$$

$\gamma$ quantifies the consistency of the inferred intention $f_n^o$ within the task context provided by the reward signal. It enables adaptation during different stages of learning and task execution: (i) *Learning Progress Adaptation*. Early in training, when the intention model $f_n^o$ is inaccurate due to partial observability and limited experience, the resulting low mutual information $\gamma$ automatically down-weights the influence of these unreliable inferences on the calibration of $\mu_\Psi$; (ii) *Task Progress Adaptation*. The factor $\gamma$ also captures the alignment between inferred intentions and the underlying task dynamics (cooperative/adversarial); even if the intention modeling (e.g., predict state) is accurate, failing to comprehend the task context can lead to actions yielding unexpected rewards $r_{t-1}^e$ and result in a low $\gamma$ as well.

Yet, the mutual information $I([r_{t-1}, f_{n-1}^o]; f_n^o)$ remains intractable. To address this issue, we draw inspiration from contrastive learning, which effectively utilizes negative sampling and acts as a regularizer, preventing collapsed solutions and enhancing the stability of the calibration process. Specifically, we employ the InfoNCE loss [39] to establish tractable bounds for the mutual information:

$$I_{\text{nce}}^c \leq I([r_{t-1}, f_{n-1}^o]; f_n^o) \leq I_{\text{nce}}^{1-c}. \tag{10a}$$

$$I_{\text{nce}}^c \triangleq \mathbb{E}_{p(x_t, s_{t+1})} \mathbb{E}_S - \log \frac{\exp(c([r_{t-1}, f_{n-1}^o], f_n^o))}{\sum_{f_i^o \in \mathcal{F}^- \cup f_n^o} \exp(c([r_{t-1}, f_{n-1}^o], f_i^o))}. \tag{10b}$$

where $c$ is a score function (implemented as bilinear+softmax in our work) assigning high scores to positive pairs $(f_{n-1}^o, r_{t-1}^e, f_n^o)$ and $\mathcal{F}^-$ denotes the set of negative examples $\{(f_{n-1}^o, r_{t-1}^e, f_j^o)\}_{j=1}^{|\mathcal{F}|}$. During learning, the positive samples are generated from the memory module, while the negative ones $f_j^o$ are computed by adding additional noise to $f_n^o$. The derivation of these bounds is detailed in Appendix C. For consistency with the inequality directions in the chance constraints, we substitute the left-hand side of Eq. (10a) into Eq. (7), while substituting the right-hand side into Eq. (8).

---

**Algorithm 1** CERMIC

---

1: **Initialize:** CERMIC and the actor-critic network
2: **for** episode $j = 1$ to $M$ **do**
3:     **for** timestep $t = 0$ to $T - 1$ **do**
4:         **for** each agent $n = 0$ to $N$ **do**
5:             Obtain action from the actor $a_t \sim \pi(s_t)$, then obtain the state $s_{t+1}$;
6:             Add $(s_t, a_t, s_{t+1})$ into the on-policy experiences;
7:             Obtain the intrinsic reward $r_t^i$ by Eq. (12) and update $r_t^e$ with $r_t = r_t^i + r_t^e$;
8:         **end for**
9:     **end for**
10:    Update the actor and critic with the collected on-policy experiences as the input;
11:    Update CERMIC by gradient descent based on Eq. (11) with the collected on-policy experiences;
12: **end for**

---

## 4.3 Loss Module

The final loss for training CERMIC is a combination of the upper and lower bounds established in previous sections. CERMIC's parameters $\Theta$ encompass the components illustrated in Fig. 1. Owing to its operation in a low-dimensional space and reliance on inputs that are readily available in standard MARL pipelines, CERMIC serves as an efficient, plug-and-play module.

$$\min_\Theta \mathcal{L}_C = L_{\text{exploit}}^{\text{UB}} + \mathcal{L}_{\text{exploit}}^{\text{LB}} - \alpha \mathcal{L}_{\text{explore}} . \tag{11}$$

## 4.4 Intrinsic Reward Module

In this section, we detail how CERMIC generates intrinsic rewards to foster exploration. Our aim is to not only encourage novelty seeking but also to be theoretically compatible with extrinsic rewards. To this end, drawing inspiration from Bayesian Surprise [24], we formulate the intrinsic reward $r_t^i$:

$$r_t^i \triangleq \mathcal{D}_{KL}(p(\Theta|(s_t, a_t, s_{t+1}) \cup \mathcal{D}_m) \| p(\Theta|\mathcal{D}_m))^{1/2}. \tag{12}$$

where $\Theta$ represents the parameters of the CERMIC module and the dataset $\mathcal{D}_m$ comprises transition tuples $(s_t, a_t, s_{t+1})$ collected over the past $m$ episodes. Intuitively, this intrinsic reward encourages agents to prioritize the exploration that are maximally informative for optimizing the CERMIC module itself. In summarize, we show the overall MARL algorithm with CERMIC in Algorithm 1.

**Theoretical Analysis in Linear MDPs** Analyzing dynamics of intrinsic rewards in high-dimensional, non-linear environments presents significant theoretical challenges. However, we provide a theoretical analysis within the framework of linear Markov Decision Processes (MDPs), where the transition kernel and reward model are assumed to be linear (i.e., $x_t = \omega_t \eta(s_t, a_t)$, where $\omega_t$ is the CERMIC parameters in linear MDP settings and $\eta$ is a feature embedding function). Within this well-studied setting, algorithms like LSVI-UCB [14] are known to achieve near-optimal worst-case regret bounds, largely due to their principled exploration strategy.

The exploration bonus in LSVI-UCB, denoted $r_t^{\text{UCB-DB}}$, quantifies the uncertainty in the value estimate for the state-action pair $(s_t, a_t)$ and encourages optimistic exploration. We proved that, Our proposed intrinsic reward $r_t^i$ is closely related to this UCB-DB bonus, as formalized in Theorem 1.

**Theorem 1** *Consider a linear MDP setting where the estimation noises of optimal parameters and transition dynamics are assumed to follow standard Gaussian distributions $\mathcal{N}(0, \mathbf{I})$. Then, for any tuning parameter $\rho > 0$, it holds that*

$$\rho \cdot r_t^{UCB\text{-}DB} \leq r_t^i \leq \sqrt{2}\rho \cdot r_t^{UCB\text{-}DB}. \tag{13}$$

By tracking the UCB-DB bonus, which naturally decays as uncertainty about state-action values diminishes, $r_t^i$ also attenuates over time. This prevents persistent, potentially destabilizing intrinsic motivation when exploration is no longer paramount, facilitating convergence towards policies optimized for extrinsic rewards.

While the intrinsic reward may be challenging due to the non-Bayesian parameterization of the CERMIC model, We empirically approximate it with an IB-trained representation and a Gaussian marginal,

yielding a tractable lower-bound bonus that supports robust exploration in noisy environments as follows:

While directly computing $r^i$ is challenging, we address this by deriving a computable lower bound using the Data Processing Inequality (DPI), which states that post-processing cannot increase information. Applying a learned representation function $\mathcal{R}(s_t, a_t)$ (a part of CERMIC) to the transition $(s_t, a_t, S_{t+1})$, the DPI yields:

$$
\begin{aligned}
r^i(s_t, a_t) &= \left[ \mathcal{H}\big((s_t, a_t, s_{t+1})|\mathcal{D}_m\big) - \mathcal{H}\big((s_t, a_t, s_{t+1})|\Theta, \mathcal{D}_m\big) \right]^{1/2} \\
&\geq \left[ \mathcal{H}\big(\mathcal{R}(s_t, a_t, s_{t+1})|\mathcal{D}_m\big) - \mathcal{H}\big(\mathcal{R}(s_t, a_t, s_{t+1})|\Theta, \mathcal{D}_m\big) \right]^{1/2} \triangleq r^i_{\text{approx}}.
\end{aligned}
\tag{14}
$$

The approximated reward $r^i_{\text{approx}}$ thus becomes the KL divergence between the conditional representation $\mathcal{R}(x_t|s_t, a_t)$ and its marginal $p_{\text{margin}}(x_t|\mathcal{D}_m)$:

$$
r^i_{\text{approx}}(s_t, a_t) = \mathbb{E}_\Theta \left[ D_{\text{KL}} \left( \mathcal{R}_\phi(x_t|s_t, a_t) \| p_{\text{margin}}(x_t|\mathcal{D}_m) \right) \right]^{1/2}.
\tag{15}
$$

The marginal $p_{\text{margin}}(x_t|\mathcal{D}_m)$ over model parameters is intractable. Following common practice in information-theoretic exploration with non-Bayesian models, we approximate $p_{\text{margin}}(x_t|\mathcal{D}_m)$ with a fixed standard Gaussian distribution $\mathcal{N}(0, \mathbf{I})$. This renders Eq. (15) empirically computable.

## 5 Experiments

### 5.1 Task Setup

**Benchmarks and Evaluation Metrics.** We evaluate our approach on a diverse set of MARL benchmarks: VMAS (9 tasks) [3], MeltingPot (4 tasks)[1], and SMACv2 (2 tasks) [9]. All environments were adapted to sparse-reward configurations to rigorously test exploration capabilities; specific modifications and implementation details are provided in Appendix E. Different metrics for each benchmark: (i) mean episodic reward for VMAS, (ii) mean episodic return for MeltingPot, and (iii) mean test win-rate for SMACv2.

**Baselines.** We integrate CERMIC with two widely-used MARL algorithms, MAPPO[15] and QMIX [31]. Our approach is compared against three categories of algorithms: (i) Standard MARL baselines: MAPPO, QMIX. (we also include a naive exploration method MAPPO-$\epsilon$ GREEDY) (ii) State-of-the-art MARL methods: CPM [19] from VMAS and QMIX-SPECTRA [28] from SMACv2. (iii) Other curiosity-driven exploration algorithms: MAPPO-DB [2], MACE [13], and ICES [20]. Crucially, all compared algorithms are required to operate under communication-limited, decentralized execution settings to ensure fair and relevant comparisons.

### 5.2 Comparison with State-of-the-Art

We present a comparative performance analysis of CERMIC-augmented algorithms against various baselines in Table 1. A general observation from these results is that curiosity-driven approaches, on average, tend to outperform traditional MARL methods. Building upon this, CERMIC consis-

Table 2: **Per-agent contributions** in Balance*.

| # Agents | 2 | 4 | 6 | 8 |
|---|---|---|---|---|
| MAPPO-DB | 12.3 | 13.7 | 13.6 | 12.1 |
| MAPPO-CERMIC | **14.4** | **16.7** | **16.4** | **17.3** |

tently enhances its base algorithms, achieving new SoTA performance on 12 out of the 16 evaluated scenarios. Furthermore, CERMIC shows its strongest gains on MeltingPot, where rewards depend on emergent, dynamic inter-agent interactions rather than fixed rules. In such settings, intention-aware exploration and curiosity calibration give a clear advantage over simple novelty seeking.

Additionally, Agents driven by single-agent curiosity formulations (e.g., MAPPO-DB), which ignore peers' intentions, are persistently distracted by others' behaviors even in later training (see Table 2). Our contextual calibration explicitly addresses this issue by transforming socially induced randomness into a learnable signal. To further validate this, we evaluate how per-agent contributions change with the number of agents in the Balance* task. Results show that MAPPO-DB's per-agent contribution decreases as agent count increases, while CERMIC mitigates this degradation:

Table 1: **Performance comparison**: against baselines, SoTA, and other curiosity-driven methods. An * indicates environments adapted to sparse-reward configurations; others are originally sparse. Task names omitted (see appendix for details)

| | VMAS | | | | | | | | | MeltingPot | | | | SMACv2 | |
|---|---|---|---|---|---|---|---|---|---|---|---|---|---|---|---|
| | Disper | Naviga | Sampli* | Passag* | Transp* | Balanc* | GiveWa* | Wheel* | Flocki* | StaHun | CleaUp | ChiGam | PriDil | pro5v5* | zer5v5* |
| *Baseline Methods* | | | | | | | | | | | | | | | |
| QMIX [32] | 0.69 | 0.65 | 25.4 | 154 | 0.42 | 48.5 | 3.40 | -2.83 | 0.55 | 5.11 | 71.2 | 6.83 | 4.57 | 0.60 | 0.39 |
| MAPPO [15] | 1.44 | 1.08 | 21.8 | 159 | 0.23 | 60.3 | 3.71 | -3.70 | -0.37 | 5.02 | 74.2 | 8.44 | 4.93 | 0.61 | 0.44 |
| MAPPO-$\epsilon$ GREEDY | 1.51 | 1.22 | 22.8 | 161 | 0.27 | 63.3 | 3.74 | -3.53 | -0.12 | 5.33 | 74.5 | 8.62 | 5.17 | 0.61 | 0.43 |
| *Current SoTA* | | | | | | | | | | | | | | | |
| CPM [19] | 1.46 | 1.25 | 25.7 | 162 | 0.44 | 61.0 | 3.73 | -2.51 | 0.20 | 5.11 | 74.4 | 8.35 | 5.21 | 0.64 | 0.44 |
| QMIX-SPECTRA [28] | 1.52 | 1.14 | 24.2 | 154 | 0.47 | 52.9 | 3.82 | -2.84 | 0.02 | 5.75 | 70.4 | 8.51 | 5.17 | 0.65 | 0.45 |
| *Curiosity Methods* | | | | | | | | | | | | | | | |
| MAPPO-DB [2] | 1.34 | 0.88 | 23.0 | 155 | 0.51 | 55.2 | 3.82 | -2.05 | -0.04 | 5.08 | 71.2 | 8.47 | 4.99 | 0.63 | 0.43 |
| MACE [13] | 1.48 | 1.22 | **28.1** | 166 | 0.60 | 60.0 | 3.62 | -1.62 | 0.24 | 6.18 | 75.2 | 8.94 | 5.52 | 0.67 | 0.44 |
| QPLEX-ICES [20] | 1.56 | 1.36 | 25.5 | 164 | 0.61 | 63.2 | **3.96** | -1.44 | 0.63 | 7.20 | 76.7 | 9.07 | 5.42 | **0.74** | 0.48 |
| **QMIX-CERMIC (ours)** | 1.02 | 0.92 | 27.2 | 163 | **0.84** | 62.7 | 3.77 | **-1.31** | 0.78 | **8.43** | 76.2 | 8.47 | 5.02 | 0.73 | 0.44 |
| **MAPPO-CERMIC (ours)** | **1.57** | **1.44** | 25.4 | **172** | 0.64 | **67.3** | 3.94 | -1.47 | **1.06** | 7.11 | **78.3** | **10.03** | **6.74** | 0.70 | **0.48** |

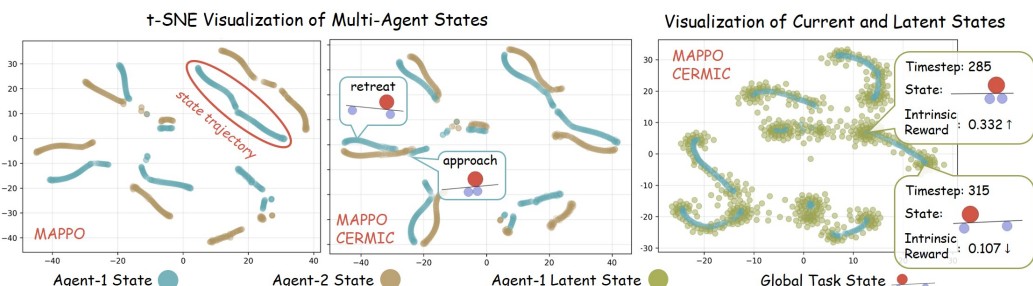

Figure 2: **Visualization of agent observation embeddings** $s_t$ **and latent states** $x_t$. (**Left**) Comparative $s_t$ distributions for agents under CERMIC-augmented vs. baseline algorithms. (**Right**) Influence of curiosity-driven latent states $x_t$ on task exploration.

## 5.3 Qualitative and Quantitative Analysis

**Qualitative Analysis.** To provide intuitive insights of the operational dynamics, we visualize the temporal evolution of agent states $s_t$ and their corresponding curiosity latent variables $x_t$ (Fig. 2). We define a *state trajectory* as a temporally contiguous sequence of $s_t$ embeddings.

(i) *Curiosity to Peers*: The left panel of Fig. 2, which displays these state trajectories, reveals that CERMIC-augmented agents exhibit more frequent close proximities and intersections in their pathways compared to those under vanilla MAPPO. These convergences, without full trajectory overlap, suggest that agents actively approach others for informed observation to refine their distinct plans, rather than engaging in mere imitation.

(ii) *Curiosity to Environment*: In the right panel, the visualization of latent states offers further compelling evidence. *Firstly*, $x_t$ often displays a more dispersed distribution around the current $s_t$ embedding, reflecting an intrinsic drive towards novel or uncertain aspects of the current state. *Secondly*, these latent states tend to concentrate at the endpoints of the state trajectories, indicating CERMIC's heightened activity prior to significant behavioral shifts, aligning with the original motivation behind designing an intrinsic reward that prioritizes critical transitions. *Furthermore*, inter-agent encounters typically trigger a surge in the intrinsic reward, indicating that CERMIC effectively captures and values the increased novelty and learning opportunities presented by these crucial multi-agent contextual shifts.

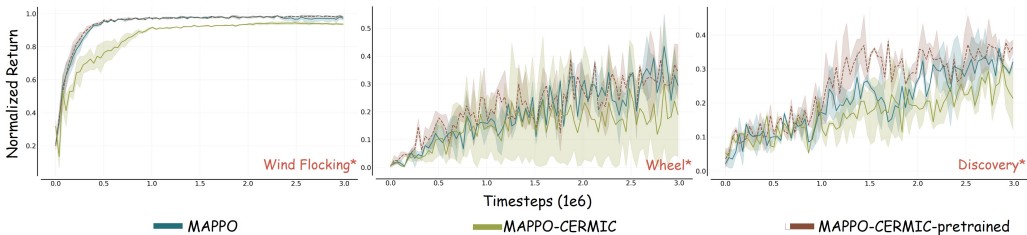

Figure 3: The impact of pretrained intention memory module on CERMIC's performances

**Quantitative Analysis.** Fig. 3 illustrates the mean episodic return curves on the VMAS task, with comprehensive results across additional tasks deferred to Appendix F. While CERMIC-augmented algorithms ultimately achieve superior final performance, their learning curves can exhibit a slower initial ramp-up compared to some baselines. We attribute this initial lower sample efficiency to the inherent challenge of learning effective multi-agent intention modeling from scratch, a difficulty induced by severe partial observability.

### 5.4 Ablation Study

We conduct three ablations on CERMIC, summarized in Table 3: (i) *Loss module*: Ablating the exploration loss $\mathcal{L}_{\text{explore}}$ markedly impedes adaptation in sparse-reward settings, yielding slower learning; removing the exploitation losses $\mathcal{L}_{\text{exploit}}^{\text{UB}} + \mathcal{L}_{\text{exploit}}^{\text{LB}}$ prevents effective noise filtering, leading to instability and pronounced fluctuations. (ii) $\alpha$: increasing the curiosity scale $\alpha$ (for our calibrated intrinsic curiosity) typically lowers immediate returns via greater exploration, yet improves learning stability and reduces performance variance, reflecting an exploration–exploitation trade-off. (iii) *Intention model*: More expressive intention models that better extract contextual features $f_n^o$ consistently improve task performance.

Table 3: **Ablation studies and hyperparameter analysis** of CERMIC.

|  | Flocki* | Naviga | Passag* | CleanUp | ChiGam |
|---|---|---|---|---|---|
| ***Loss Ablations*** ($\alpha = 0.2$) | | | | | |
| w/o $L_{\text{explore}}$ | 0.82 ($\pm$0.15) | 1.41 ($\pm$0.04) | 166 ($\pm$3.66) | 75.4 ($\pm$1.05) | 9.22 ($\pm$0.80) |
| w/o $L_{\text{exploit}}$ | 0.72 ($\pm$0.48) | 1.36 ($\pm$0.10) | 164 ($\pm$4.41) | 71.5 ($\pm$1.59) | 8.47 ($\pm$1.18) |
| MAPPO-CERMIC | 1.06 ($\pm$0.12) | 1.43 ($\pm$0.06) | 171 ($\pm$3.72) | 78.1 ($\pm$0.82) | 10.02 ($\pm$0.65) |
| ***Coverage Level*** $\alpha$ | | | | | |
| 1.0 | 0.80 ($\pm$0.07) | 1.40 ($\pm$0.02) | 163 ($\pm$1.56) | 71.2 ($\pm$0.77) | 9.04 ($\pm$0.46) |
| 0.5 | 0.88 ($\pm$0.10) | 1.42 ($\pm$0.06) | 169 ($\pm$3.78) | 74.6 ($\pm$0.68) | 9.50 ($\pm$0.67) |
| 0.2 | 1.06 ($\pm$0.12) | 1.43 ($\pm$0.06) | 171 ($\pm$3.72) | 78.1 ($\pm$0.82) | 10.02 ($\pm$0.65) |
| ***Memory Type*** ($\alpha = 0.2$) | | | | | |
| GRU | 0.78 ($\pm$0.27) | 1.39 ($\pm$0.04) | 161 ($\pm$3.15) | 75.2 ($\pm$1.03) | 9.19 ($\pm$0.97) |
| Graph | 1.06 ($\pm$0.12) | 1.43 ($\pm$0.06) | 171 ($\pm$3.72) | 78.1 ($\pm$0.82) | 10.02 ($\pm$0.65) |

### 5.5 Generalization Across Reward Densities

To validate this and demonstrate CERMIC's potential with proficient intention modeling, we conducted an auxiliary experiment: We use pre-trained agent detection and intention modeling modules to provide more accurate initial estimates of other agents' states, leading to a CERMIC variant that exhibited significantly faster convergence. Given advancements in related areas like motion prediction, this result underscores CERMIC's strong applicability.

We further investigated the generalization capabilities of agents by evaluating their performance when trained under one reward density (dense or sparse) and tested under a sparse-reward setting. The results are presented in Table 4. A consistent trend observed across all algorithms is a performance degradation when agents trained on dense rewards are deployed in sparse-reward scenarios. However, CERMIC demonstrates a notable ability to mitigate this performance drop. We attribute this to CERMIC's capability to enable agents to understand the task by observing other agents, rather than

solely relying on trial-and-error learning. These findings underscore the importance of research in sparse-reward reinforcement learning and highlight the potential for CERMIC's broad applicability.

Table 4: **Agent generalization performance on** VMAS **tasks**. $\Delta$ denotes the performance drop from dense-trained to sparse-trained.

| Method | Balance* | | | Give Way* | | | Passage* | | |
|---|---|---|---|---|---|---|---|---|---|
| | Sparse↑ | Dense↑ | $\Delta$↓ | Sparse↑ | Dense↑ | $\Delta$ | Sparse↑ | Dense↑ | $\Delta$↓ |
| CPM | 61.0 | 58.2 | -2.8 | 3.73 | 3.58 | -0.15 | 162 | 153 | -9 |
| QPLEX-ICES | 63.2 | 60.7 | -2.5 | **3.96** | **3.85** | **-0.11** | 164 | 159 | -5 |
| **MAPPO-CERMIC** | **67.3** | **65.6** | **-1.7** | 3.94 | 3.82 | -0.12 | **172** | **168** | **-4** |

## 6 Conclusion

This paper introduced CERMIC, a novel plug-and-play module significantly enhancing exploration in communication-limited, partially observable MARL under sparse rewards. CERMIC's core innovation is a multi-agent contextualized calibration of intrinsic curiosity. Grounded in Bayesian Surprise and supported by theoretical guarantees, we also propose a novel intrinsic reward guides this calibrated exploration. Extensive experiments show CERMIC-augmented algorithms achieve state-of-the-art performance, underscoring the efficacy of context-aware intrinsic motivation. In essence, CERMIC offers a crucial step towards enabling more autonomous and socially intelligent agent teams for real-world deployment. No negative societal impact.

**Limitations and Future Work.** Despite its promising results, CERMIC has limitations that open avenues for future research. *First*, learning to accurately model other agents' intentions solely from local observations in communication-less, partially observable settings remains a challenging endeavor. Future work could explore integrating pre-trained components, such as LLMs, to potentially bootstrap or replace aspects of this ad-hoc intention modeling process. *Second*, our current graph construction and one-step latent transition model may require extensions (e.g., hierarchical latent models) to effectively scale to higher-dimensional or multi-modal MAS tasks.

## Acknowledgment

This paper was supported by the Natural Science Foundation of China under Grant 62303307, 62225309, U24A20278, 62361166632, U21A20480, and Sponsored by the Oceanic Interdisciplinary Program of Shanghai Jiao Tong University, and in part by National Key Laboratory of Human Machine Hybrid Augmented Intelligence, Xi'an Jiaotong University (No. HMHAI-202408).

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

# A  Memory Update Module

The agent memory update module primarily consists of an agent detection mechanism and a memory encoding structure. In our main approach, this involves an MLP-based agent detector and a graph-based memory module. This section elaborates on the graph structure and the pre-training of the detector.

## A.1  Graph-based Memory Representation

The graph-based memory module offers a structured approach to model inter-agent states and their relationships. For each observing agent, a local graph $\mathcal{G}_t = (\mathcal{V}_t, \mathcal{E}_t)$ is dynamically constructed or updated at each timestep $t$.

- **Nodes ($\mathcal{V}_t$):** Each node $v_i \in \mathcal{V}_t$ corresponds to an inferred latent state representation of a specific agent $i$ (either the self-agent or a detected peer). These node features, denoted $\mathbf{z}_i \in \mathbb{R}^{D_{node}}$, are generated by encoding the raw observation associated with agent $i$.
- **Edges ($\mathcal{E}_t$):** Edges $e_{ij} \in \mathcal{E}_t$ link pairs of nodes $(v_i, v_j)$, representing the inferred relationship between agent $i$ and agent $j$. In our work, these are spatial relationships, with edge features $\mathbf{f}_{ij} \in \mathbb{R}^{D_{edge}}$ derived from their relative positions.

This dynamic graph serves as input to a Graph Neural Network (GNN), which processes the relational information to produce a contextualized memory representation for the observing agent, thereby informing its subsequent actions.

## A.2  Pre-training Process

Key components of the agent memory module are pre-trained to establish meaningful initial representations, facilitating more effective subsequent MARL training.

**MLP Agent Detector.**     This module processes an agent's observation to output probabilities $\mathbf{p} = [p_1, \ldots, p_N]^\top$, where $p_k$ denotes the likelihood of observing an agent of type-$k$. These probabilities are later used to gate memory updates during MARL training via a threshold $\tau_{\text{det}}$. Pre-training is supervised using ground-truth presence labels $y_k \in \{0, 1\}$, which are set to 1 if agent-$k$ appears within the field of view or one body length of the observer, and 0 otherwise. To facilitate reliable detection, agents in our environment are either designed with or inherently exhibit distinct visual appearances, and each agent type is represented by a dedicated channel in the observation tensor, enabling effective training of an MLP-based detector.

$$\mathcal{L}_{det} = -\sum_{k=1}^{N} \left[ y_k \log(p_k) + (1 - y_k) \log(1 - p_k) \right]. \tag{16}$$

**Graph Memory Encoders.**  The node and edge encoders within the graph memory are pre-trained as follows: (i) *Node Encoder*: for an observing agent $k$, its encoder produces a feature $\mathbf{z}_{k \to j}$ representing its understanding of another agent $j$'s state. This is trained to predict agent $j$'s true latent state $\mathbf{s}_j$. The MSE loss is weighted by a detection mask, which is derived from the predicted transition probabilities $p_{k \to j}$ produced by a pre-trained detector, using a fixed threshold to determine valid entries; (ii) *Edge Encoder*: this encoder generates 2D edge features $\mathbf{f}_{ij}$ from inter-agent cues, trained to directly predict the true relative position vector $(\Delta x_{ij}, \Delta y_{ij})$ between agents $i$ and $j$ using an MSE loss.

$$\begin{cases} \mathcal{L}_{node}^{(k)} = \sum_{j \neq k} p_{k \to j} \cdot \mathbb{E} \left[ ||\mathbf{z}_{k \to j} - \mathbf{s}_j||^2 \right]. \\ \mathcal{L}_{edge} = \mathbb{E} \left[ ||\mathbf{f}_{ij} - (\Delta x_{ij}, \Delta y_{ij})||^2 \right]. \end{cases} \tag{17}$$

These pre-training steps provide the memory module with an initial capacity for relevant information extraction, potentially accelerating overall learning.

# B  Derivation with Cantelli Inequality

In this section, we detail the transformation of the distributionally robust chance constraint, Eq.(6) of the main paper. This transformation leverages Cantelli's inequality to arrive at a tractable loss

under second-order cone program (SOCP) constraint. In this section, we set $\bar{c} = 1$ for illustration. We begin by defining auxiliary variables based on the quantities in the chance constraint:

$$\tilde{s} = \Psi_t - \mu_\Psi(f_n^o) \tag{18a}$$

$$b = 1 - \mu_\Psi(f_n^o) \tag{18b}$$

We also define the set $S$ describing the ambiguity in the first and second moments of $\tilde{s}$, and the set $\mathcal{D}_{\tilde{s}}$ of distributions consistent with these moments:

$$S = \left\{ (\mu_1, \sigma_1) : |\mu_1| \leq \sqrt{\gamma_1 \Sigma_\Psi}, \mu_1^2 + \sigma_1^2 \leq \gamma_2 \Sigma_\Psi \right\} \tag{19a}$$

$$\mathcal{D}_{\tilde{s}} = \left\{ \mathbb{P}_r \in \mathcal{P}(\mathbb{R}) : \begin{array}{l} |\mathbb{E}_{\mathbb{P}_r}[\tilde{s}]| \leq \sqrt{\gamma_1 \Sigma_\Psi} \\ \mathbb{E}_{\mathbb{P}_r}[\tilde{s}^2] \leq \gamma_2 \Sigma_\Psi \end{array} \right\} \tag{19b}$$

where $\mathcal{P}(\mathbb{R})$ is the set of all probability distributions on $\mathbb{R}$. Then, the original chance constraint $\inf_{\mathbb{P}_r \in \mathcal{P}_{\text{cal}}} \mathbb{P}_r(\Psi_t \leq 1) \geq 1 - \epsilon$ can be rewritten using $\tilde{s}$ and $b$. The infimum term is:

$$\begin{aligned} \inf_{\mathbb{P}_r \in \mathcal{P}_{\text{cal}}} \mathbb{P}_r(\Psi_t \leq 1) &= \inf_{\mathbb{P}_r \in \mathcal{D}_{\tilde{s}}} \mathbb{P}_r(\tilde{s} \leq b) \\ &= \inf_{(\mu_1, \sigma_1) \in S} \inf_{\mathbb{P}_r \in \mathcal{P}(\mu_1, \sigma_1^2)} \mathbb{P}_r(\tilde{s} \leq b). \end{aligned} \tag{20}$$

where $\mathcal{P}_{\text{cal}}$ refers to the set of distributions under multi-agent contextual intention uncertainty in the main paper. To this end, Eq. (20) formulates the problem as a bi-level optimization. The outer layer finds the worst-case mean $\mu_1$ and variance $\sigma_1^2$ within the ambiguity set $S$. The inner layer finds the worst-case probability $\mathbb{P}_r(\tilde{s} \leq b)$ for a given mean and variance.

To solve the inner optimization problem, we employ the one-sided Cantelli's inequality. For a random variable $X$ with mean $\mathbb{E}[X]$ and variance $\text{Var}[X] = \sigma^2$, Cantelli's inequality provides bounds on tail probabilities. Specifically, for the lower tail, the bound is defined as:

$$\mathbb{P}_r(X - \mathbb{E}(X) \geq -\lambda) \leq \frac{\sigma^2}{\sigma^2 + \lambda^2}$$

Applying this to our inner problem $\inf_{\mathbb{P}_r \in \mathcal{P}(\mu_1, \sigma_1^2)} \mathbb{P}_r(\tilde{s} \leq b)$, we get:

$$\inf_{\mathbb{P}_r \in \mathcal{P}(\mu_1, \sigma_1^2)} \mathbb{P}_r(\tilde{s} \leq b) = \begin{cases} \frac{(b - \mu_1)^2}{\sigma_1^2 + (b - \mu_1)^2}, & \text{if } b \geq \mu_1 \\ 0, & \text{otherwise.} \end{cases} \tag{21}$$

In practical multi-agent reinforcement learning scenarios, the desired confidence level $1 - \epsilon$ is positive. For the chance constraint $\inf \mathbb{P}_r(\tilde{s} \leq b) \geq 1 - \epsilon$ to hold with $1 - \epsilon > 0$, we must be in the regime where $b \geq \mu_1$. If $b < \mu_1$, the infimum probability would be 0, violating the constraint. Therefore, for any relevant $(\mu_1, \sigma_1) \in S$, the condition $b \geq \mu_1$ must be satisfied. This implies $b \geq \sup_{(\mu_1, \sigma_1) \in S} \mu_1 = \sqrt{\gamma_1 \Sigma_h}$ (assuming $\Sigma_h > 0$). Consequently, Eq. (20) simplifies to:

$$\inf_{\mathbb{P}_r \in \mathcal{P}_{\text{cal}}} \mathbb{P}_r(\Psi_t \leq 1) = \inf_{(\mu_1, \sigma_1) \in S} \frac{(b - \mu_1)^2}{\sigma_1^2 + (b - \mu_1)^2} \tag{22}$$

Solving the optimization problem in Eq. (22) over the set $S$ (defined by moment bounds $|\mu_1| \leq \sqrt{\gamma_1 \Sigma_h}$ and $\mu_1^2 + \sigma_1^2 \leq \gamma_2 \Sigma_h$) yields the following worst-case probability:

$$\inf_{\mathbb{P}_r \in \mathcal{P}_{\text{cal}}} \mathbb{P}_r(\Psi_t \leq 1) = \begin{cases} \frac{1}{\left( \frac{\sqrt{\gamma_2 - \gamma_1}}{b/\sqrt{\Sigma_h} - \sqrt{\gamma_1}} \right)^2 + 1}, & \text{if } \sqrt{\gamma_1} \leq \frac{b}{\sqrt{\Sigma_\Psi}} \leq \frac{\gamma_2}{\sqrt{\gamma_1}} \\ \frac{(b/\sqrt{\Sigma_h})^2 - \gamma_2}{(b/\sqrt{\Sigma_h})^2}, & \text{if } \frac{b}{\sqrt{\Sigma_\Psi}} > \frac{\gamma_2}{\sqrt{\gamma_1}} \\ \text{infeasible,} & \text{if } \frac{b}{\sqrt{\Sigma_\Psi}} < \sqrt{\gamma_1}. \end{cases}$$

The chance constraint requires this infimum probability to be at least $1 - \epsilon$: $\inf_{\mathbb{P}_r \in \mathcal{P}_{\text{cal}}} \mathbb{P}_r(\Psi_t \leq 1) \geq 1 - \epsilon$. By substituting the expressions above and performing algebraic manipulations, we arrive at the tractable second-order cone constraint. Substituting $b = 1 - \Psi_t(f_n^o)$, the condition becomes:

$$\mu_\Psi(f_n^o) + \beta \sqrt{\Sigma_\Psi} \leq 1 \tag{23a}$$

$$l_n = \begin{cases} \sqrt{\gamma_1} + \sqrt{\frac{1 - \epsilon}{\epsilon}(\gamma_2 - \gamma_1)}, & \text{if } \gamma_1/\gamma_2 \leq \epsilon \leq 1 \\ \sqrt{\frac{\gamma_2}{\epsilon}}, & \text{if } 0 < \epsilon < \gamma_1/\gamma_2. \end{cases} \tag{23b}$$

This provides the final tractable form for the first inequality in Eq. (6) of the main paper. For this second-order cone constraint, we wrap it with a ReLU function to construct the following loss $\mathcal{L}_{\text{exploit}}^{\text{UB}}$, which forms an upper bound of the exploit loss. We substitute $\bar{c}$ to recover the Eq. (7) presented in the main text.

To ensure stability and prevent variables $\Psi_t$ from shrinking indefinitely, we also impose a lower-bound constraint $\mathbb{P}_r(\Psi_t \geq \underline{c}) \geq 1 - \epsilon$, which can be converted into loss $\mathcal{L}_{\text{exploit}}^{\text{LB}}$ in a similar manner.

## C   InfoNCE Bounds on Mutual Information

This appendix details how the InfoNCE objective provides lower and upper bounds for the mutual information $I([r_{t-1}^e, f_{n-1}^o]; f_n^o)$. We begin our derivation by establishing the lower bound relationship $I_{\text{nce}}(c) \leq I([r_{t-1}^e, f_{n-1}^o]; f_n^o)$. For notational simplicity, we denote the context $[r_{t-1}^e, f_{n-1}^o]$ as $z_n$.

In our work, the InfoNCE objective is defined using a critic function $c([r_{t-1}^e, f_{n-1}^o], f_n^o)$, which measures the compatibility between the context $[r_{t-1}^e, f_{n-1}^o]$ and the current inferred intention $f_n^o$. In practice, $c(\cdot, \cdot)$ is implemented as a bilinear layer followed by a softmax, yielding scores in the range $[0, 1]$. Given a positive pair $(f_n^o, z_n)$ drawn from the joint distribution $p(z_n, f_n^o)$, and a set of $N$ negative samples $\mathcal{F}^- = \{f_n^{-o,j}\}_{j=1}^N$ drawn from the complement of $f_n^o$ in its state space $\mathcal{F}$, the InfoNCE objective is defined as:

$$I_{\text{nce}}^c \triangleq \mathbb{E}_{p(z_n, f_n^o)} \mathbb{E}_{\mathcal{F}^-} \left[ \log \frac{\exp(c(z_n, f_n^o))}{\sum_{j=1}^N \exp(c(z_n, f_n^{-o,j}))} \right]. \tag{24}$$

where the positive pairs $(z_n, f_n^o)$ are obtained from the current prediction pair. Negative samples are generated by independently sampling $N$ times from the state space of $f_n^o$. Let $\mathbb{I}$ be an indicator variable, with $\mathbb{I} = 1$ indicating a positive sample randomly drawn from the state space, and $\mathbb{I} = 0$ indicating a negative one (i.e., $f_n^o$ is replaced by $f_n^{-o}$). The corresponding conditional probabilities have the following properties:

$$\begin{cases} p(z_n, f_n^o | \mathbb{I} = 1) = p(z_n, f_n^o). \\ p(z_n, f_n^{-o} | \mathbb{I} = 0) = p(z_n)p(f_n^o). \end{cases} \tag{25}$$

We aim to maximize the InfoNCE objective $I_{\text{nce}}^c$. We assume that the optimal critic $c^*$ can identify the log-posterior probability of a randomly sampled pair being the positive one. Thus, we have:

$$\begin{aligned} I_{\text{nce}}^c &\leq I_{\text{nce}}^{c*} \\ &\leq \mathbb{E}_{p(z_n, f_n^o)} \mathbb{E}_{\mathcal{F}^-} \left[ c^*(z_n, f_n^o) \right] \\ &= \mathbb{E}_{p(z_n, f_n^o)} \mathbb{E}_{\mathcal{F}^-} \left[ \log p(\mathbb{I} = 1 | z_n, f_n^o, \mathcal{F}^-) \right]. \end{aligned} \tag{26}$$

Next, we expand $\log p(\mathbb{I} = 1 | z_n, f_n^o, \mathcal{F}^-)$ using Bayes' theorem. Considering one positive sample and $N$ negative samples, the prior probabilities are $p(\mathbb{I} = 1) = 1/(N+1)$ and $p(\mathbb{I} = 0) = N/(N+1)$ for the collection of negative samples.

$$\begin{aligned} \log p(\mathbb{I} = 1 | z_n, f_n^o, \mathcal{F}^-) &= \log \frac{p(z_n, f_n^o | \mathbb{I} = 1)p(\mathbb{I} = 1)}{p(z_n, f_n^o | \mathbb{I} = 1)p(\mathbb{I} = 1) + p(z_n, f_n^{-o} | \mathbb{I} = 0)p(\mathbb{I} = 0)} \\ &= \log \frac{p(z_n, f_n^o)}{p(z_n, f_n^o) + Np(z_n)p(f_n^o)}. \end{aligned} \tag{27}$$

This expression for $\log p(\mathbb{I} = 1 | z_n, f_n^o, \mathcal{F}^-)$ represents the log-probability of the given $f_n^o$ being the true positive, relative to $N$ distractors drawn from $p(z_n)p(f_n^o)$. Continuing from this expression:

$$\begin{aligned} \log \frac{p(z_n, f_n^o)}{p(z_n, f_n^o) + Np(z_n)p(f_n^o)} &= \log \left( \frac{p(z_n, f_n^o)}{p(z_n)p(f_n^o)} \cdot \frac{p(z_n)p(f_n^o)}{p(z_n, f_n^o) + Np(z_n)p(f_n^o)} \right) \\ &\leq \log \frac{p(z_n, f_n^o)}{p(z_n)p(f_n^o)} - \log N. \end{aligned} \tag{28}$$

Taking the expectation $\mathbb{E}_{p(z_n, f_n^o)} \mathbb{E}_{\mathcal{F}^-}$ on both sides of the result from Eq. (26) combined with the inequality above:

$$\mathbb{E}_{p(z_n, f_n^o)} \mathbb{E}_{\mathcal{F}^-} \left[ \log p(\mathbb{I} = 1 | z_n, f_n^o, \mathcal{F}^-) \right] \leq \mathbb{E}_{p(z_n, f_n^o)} \left[ \log \frac{p(z_n, f_n^o)}{p(z_n)p(f_n^o)} \right] - \log N. \tag{29}$$

The term $\mathbb{E}_{p(z_n, f_n^o)} \left[ \log \frac{p(z_n, f_n^o)}{p(z_n)p(f_n^o)} \right]$ is the definition of mutual information $I(z_n; f_n^o)$. Therefore, by substituting $z_n = [r_{t-1}^e, f_{n-1}^o]$ back into Eq. (26) and rewriting $I(z_n; f_n^o)$ with the original notation, we have:

$$I([r_{t-1}^e, f_{n-1}^o]; f_n^o) \geq I_{\text{nce}}^c + \log N. \tag{30}$$

This demonstrates that the mutual information is lower-bounded by the InfoNCE objective $I_{\text{nce}}^c$ plus a term $\log N$. This justifies using $I_{\text{nce}}^c$ as a tractable surrogate to maximize a lower bound on the mutual information.

Similarly, the upper bound inequality can be derived using the same method.

$$\begin{cases} I([r_{t-1}^e, f_{n-1}^o]; f_n^o) \leq I_{\text{nce}}^{1-c}. \\ I_{\text{nce}}^{1-c} \triangleq \log N - \mathbb{E}_{p(z_n, f_n^o)} \mathbb{E}_{\mathcal{F}^-} \left[ \log \frac{\exp(1 - c(z_n, f_n^o))}{\sum_{j=1}^N \exp(1 - c(z_n, f_n^{-o,j}))} \right]. \end{cases} \tag{31}$$

# D    Intrinsic Reward in Linear MDPs

This appendix provides a theoretical justification for our proposed intrinsic reward by connecting it to the exploration bonus in LSVI-UCB within the context of linear MDPs.

## D.1    Preliminary

Least-Squares Value Iteration with Upper Confidence Bounds (LSVI-UCB) is an algorithm designed for efficient exploration and learning in linear MDPs. In a linear MDP, the transition kernel and reward function are assumed to be linear with respect to a $d$-dimensional feature map $\eta(s, a)$ of state-action pairs. Consequently, for any policy $\pi$, the action-value function $Q^\pi(s, a)$ can be expressed as $\chi^\top \eta(s, a)$ for some parameter vector $\chi \in \mathbb{R}^d$.

LSVI-UCB iteratively collects data and updates the $Q$-function parameters. In each episode, the agent acts according to the current optimistic $Q$-function. The parameter $\chi_t$ is then updated via regularized least-squares:

$$\chi_t \leftarrow \arg\min_{\chi \in \mathbb{R}^d} \sum_{i=0}^m \left[ r_t(s_t^i, a_t^i) + \gamma \max_{a'} Q_t^{\text{target}}(s_{t+1}^i, a') - \chi^\top \eta(s_t^i, a_t^i) \right]^2 + \lambda \|\chi\|_2^2,$$

where $m$ is the number of collected transitions (indexed by $i$), $\lambda > 0$ is a regularization parameter, and $Q_t^{\text{target}}$ is typically a previous estimate or a slowly updating target. The closed-form solution involves the Gram matrix $\Lambda_t = \sum_{i=0}^m \eta(s_t^i, a_t^i) \eta(s_t^i, a_t^i)^\top + \lambda I$. Crucially, LSVI-UCB employs a UCB-style exploration bonus to construct an optimistic $Q$-function: $Q_t(s, a) = \chi_t^\top \eta(s, a) + r^{\text{ucb}}(s, a)$, where the bonus is $r^{\text{ucb}}(s, a) = \zeta \left[ \eta(s, a)^\top \Lambda_t^{-1} \eta(s, a) \right]^{1/2}$. This bonus quantifies the epistemic uncertainty associated with the value estimate of $(s, a)$.

## D.2    Connection to LSVI-UCB Bonus

We establish a connection between our intrinsic reward and the LSVI-UCB exploration bonus. In linear MDPs, CERMIC's parameters $\Theta$ is rewritten as $\omega_t$ and our curiosity representation $x_t$ can be represented as $x_t = \omega_t \eta(s_t, a_t) \in \mathbb{R}^c$. Here, $\eta(s_t, a_t) \in \mathbb{R}^d$ is the state-action encoding, and $\omega_t \in \mathbb{R}^{c \times d}$ is a parameter matrix. This representation is learned by predicting the next state $s_{t+1}$ via the regularized least-squares problem:

$$\omega_t \leftarrow \arg\min_W \sum_{i=0}^m \left\| s_{t+1}^i - \omega \eta(s_t^i, a_t^i) \right\|_F^2 + \lambda \|\omega\|_F^2, \tag{32}$$

The proof proceeds by vectorizing the matrix $\omega_t$ and defining an expanded feature matrix $\tilde{\eta}$. Specifically, $\text{vec}(\omega_t) \in \mathbb{R}^{cd}$ is the column-wise vectorization of $\omega_t$, and $\tilde{\eta}(s_t, a_t) \in \mathbb{R}^{cd \times c}$ is a block-diagonal matrix with $\eta(s_t, a_t)$ repeated $c$ times along its diagonal. This construction satisfies $\text{vec}(\omega_t)^\top \tilde{\eta}(s_t, a_t) = (\omega_t \eta(s_t, a_t))^\top$. For clarity, the definitions are:

$$\begin{cases} \text{vec}(\omega_t) = [w_{11}, \ldots, w_{1d}, w_{21}, \ldots, w_{cd}]^\top \in \mathbb{R}^{cd}. \\ \tilde{\eta}(s_t, a_t) = \mathbf{I}_c \otimes \eta(s_t, a_t) \in \mathbb{R}^{cd \times c}. \end{cases} \tag{33}$$

where $\mathbf{I}(\mathbf{I}_c)$ denotes the identity matrix (with $c$ dimensions), and $\otimes$ is the Kronecker product.

**Assumption 1 (Gaussian Prior)** *We consider a linear model for predicting the c-dimensional next state $s_{t+1}$ from d-dimensional state-action features $\eta(s_t, a_t)$, formulated as $s_{t+1} = W\eta(s_t, a_t) + \xi_t$. The parameter matrix $W \in \mathbb{R}^{c \times d}$ is assumed to follow a zero-mean Gaussian prior distribution $W \sim \mathcal{N}(0, \lambda^{-1}I)$, and the model noise $\xi_t \in \mathbb{R}^c$ is assumed to follow a standard multivariate Gaussian distribution $\xi_t \sim \mathcal{N}(0, I_c)$, independent of $W$ and $\eta(s_t, a_t)$.*

To analyze the intrinsic reward, we adopt a Bayesian linear regression perspective for Eq. (32). Our goal of the analysis is to obtain the posterior distribution $p(\omega_t|\mathcal{D}_m)$ to compute the Bayesian Surprise in the intrinsic reward. Firstly, under Assumption 1, we have:

$$s_{t+1}|(s_t, a_t), \omega_t \sim \mathcal{N}\left(\omega_t \eta(s_t, a_t), \mathbf{I}\right) \equiv \mathcal{N}\left(\tilde{\eta}(s_t, a_t)^\top \text{vec}(\omega_t), \mathbf{I}\right). \tag{34}$$

Given the Gaussian prior $\text{vec}(W) \sim \mathcal{N}(0, \lambda^{-1}I)$, Bayes' rule states:

$$\log p(\text{vec}(\omega_t)|\mathcal{D}_m) = \log p(\text{vec}(\omega_t)) + \sum_{i=0}^m \log p(s_{t+1}^i|s_t^i, a_t^i, \text{vec}(\omega_t)) + \text{C}. \tag{35}$$

where C is a constant. Then, substituting the Gaussian PDF for Eq. (34) into Eq. (35) yields the log-posterior:

$$\log p(\text{vec}(\omega_t)|\mathcal{D}_m) = -\frac{\lambda}{2}\|\text{vec}(\omega_t)\|_2^2 - \frac{1}{2}\sum_{i=0}^m \|\tilde{\eta}(s_t^i, a_t^i)^\top \text{vec}(\omega_t) - s_{t+1}^i\|_2^2 + \text{C}$$

$$= -\frac{1}{2}(\text{vec}(\omega_t) - \tilde{\mu}_t)^\top \tilde{\Lambda}_t(\text{vec}(\omega_t) - \tilde{\mu}_t) + \text{C}', \tag{36}$$

where C' is a constant and $\tilde{\mu}_t = \tilde{\Lambda}_t^{-1} \sum_{i=0}^m \tilde{\eta}(s_t^i, a_t^i)s_{t+1}^i$, $\tilde{\Lambda}_t = \sum_{i=0}^m \tilde{\eta}(s_t^i, a_t^i)\tilde{\eta}(s_t^i, a_t^i)^\top + \lambda \mathbf{I}$. Thus, the posterior distribution is $\text{vec}(\omega_t)|\mathcal{D}_m \sim \mathcal{N}(\tilde{\mu}_t, \tilde{\Lambda}_t^{-1})$. The covariance matrix of $\text{vec}(\omega_t)$ is $\tilde{\Lambda}_t^{-1}$. The structure of $\tilde{\Lambda}_t$ is:

$$\tilde{\Lambda}_t = \mathbf{I}_n \otimes \left(\sum_{i=0}^m \eta(s_t^i, a_t^i)\eta(s_t^i, a_t^i)^\top + \lambda\mathbf{I}\right) = \mathbf{I}_n \otimes \Lambda_t, \tag{37}$$

Notably, $\Lambda_t = \sum_{i=0}^m \eta(s_t^i, a_t^i)\eta(s_t^i, a_t^i)^\top + \lambda\mathbf{I}$ is the Gram matrix from LSVI-UCB. To this end, the intrinsic reward, related to Bayesian Surprise, can be simplified to a change in differential entropy, where $\text{Cov}(\cdot)$ denotes the covariance:

$$\begin{aligned}
&\mathcal{D}_{KL}(p(\omega_t|(s_t, a_t, s_{t+1}) \cup \mathcal{D}_m)\|p(\omega_t|\mathcal{D}_m)) \\
&= \mathcal{D}_{KL}(p(\text{vec}(\omega_t)|(s_t, a_t, s_{t+1}) \cup \mathcal{D}_m)\|p(p(\text{vec}(\omega_t)|\mathcal{D}_m)) \\
&= \mathcal{H}(\text{vec}(\omega_t)|\mathcal{D}_m) - \mathcal{H}(\text{vec}(\omega_t)|(s_t, a_t, S_{t+1}) \cup \mathcal{D}_m) \\
&= \left[\log\det\left(\text{Cov}(\text{vec}(\omega_t)|\mathcal{D}_m)\right) - \log\det\left(\text{Cov}(\text{vec}(\omega_t)|(s_t, a_t, S_{t+1}) \cup \mathcal{D}_m)\right)\right]/2.
\end{aligned} \tag{38}$$

Substituting the covariance $\tilde{\Lambda}_t^{-1}$ and the updated covariance after observing $(s_t, a_t, s_{t+1})$:

$$\begin{aligned}
&\mathcal{D}_{KL}(p(\omega_t|(s_t, a_t, s_{t+1}) \cup \mathcal{D}_m)\|p(\omega_t|\mathcal{D}_m)) \\
&= \left[\log\det\left(\tilde{\Lambda}_t + \tilde{\eta}(s_t, a_t)\tilde{\eta}(s_t, a_t)^\top\right) - \log\det\left(\tilde{\Lambda}_t\right)\right]/2 \\
&= \left[\log\det\left(\mathbf{I} + \tilde{\eta}(s_t, a_t)^\top \tilde{\Lambda}_t^{-1} \tilde{\eta}(s_t, a_t)\right)\right]/2,
\end{aligned} \tag{39}$$

where the last equality follows from the Matrix Determinant Lemma. Given the block-diagonal structure of $\tilde{\eta}(s_t, a_t)$ and $\tilde{\Lambda}_t^{-1}$, the term $\tilde{\eta}(s_t, a_t)^\top \tilde{\Lambda}_t^{-1} \tilde{\eta}(s_t, a_t)$ is a diagonal matrix with $c$ identical scalar entries $\eta(s_t, a_t)^\top \Lambda_t^{-1} \eta(s_t, a_t)$. Thus, Eq. (39) simplifies to:

$$\begin{aligned}
&\mathcal{D}_{KL}(p(\omega_t|(s_t, a_t, s_{t+1}) \cup \mathcal{D}_m)\|p(\omega_t|\mathcal{D}_m)) \\
&= \left[\log\det\left(\mathbf{I}_c \otimes \eta(s_t, a_t)^\top \Lambda_t^{-1} \eta(s_t, a_t) + \mathbf{I}_c\right)\right]/2 \\
&= (c/2) \cdot \log\left(1 + \eta(s_t, a_t)^\top \Lambda_t^{-1} \eta(s_t, a_t)\right).
\end{aligned} \tag{40}$$

By leveraging the inequality $\frac{x}{2} \leq \log(1 + x) \leq x$, we obtain:

$$\frac{\eta(s_t, a_t)^\top \Lambda_t^{-1} \eta(s_t, a_t)}{2} \leq \log\left(1 + \eta(s_t, a_t)^\top \Lambda_t^{-1} \eta(s_t, a_t)\right) \leq \eta(s_t, a_t)^\top \Lambda_t^{-1} \eta(s_t, a_t). \tag{41}$$

Combining these results, we obtain the bounds for the square root of the mutual information:

$$\sqrt{\frac{c}{4}} \left[ \eta(s_t, a_t)^\top \Lambda_t^{-1} \eta(s_t, a_t) \right]^{1/2} \leq r_t^i \leq \sqrt{\frac{c}{2}} \left[ \eta(s_t, a_t)^\top \Lambda_t^{-1} \eta(s_t, a_t) \right]^{1/2}. \tag{42}$$

Recalling that the LSVI-UCB bonus is $r^{\text{ucb}} = \zeta \left[ \eta(s_t, a_t)^\top \Lambda_t^{-1} \eta(s_t, a_t) \right]^{1/2}$, we can express the relationship as:

$$\frac{1}{\zeta} \sqrt{\frac{c}{4}} \cdot r^{\text{ucb}} \leq r_t^i \leq \frac{1}{\zeta} \sqrt{\frac{c}{2}} \cdot r^{\text{ucb}}. \tag{43}$$

Defining $\rho = \frac{1}{\zeta} \sqrt{\frac{c}{4}}$, this can be written as $\rho \cdot r^{\text{ucb}} \leq r_t^i \leq \sqrt{2} \rho \cdot r^{\text{ucb}}$. This proof establishes a connection between our proposed intrinsic reward and the UCB bonus, providing theoretical grounding for the stability of our reward mechanism.

## E    Benchmark Adaptation to Sparse-Reward Settings

Our CERMIC model and all compared algorithms were trained in environments with sparse reward signals to specifically assess exploration capabilities. However, for a comprehensive evaluation against baselines, particularly those not designed for extreme sparsity, final performance was measured using the original dense extrinsic rewards provided by the benchmarks. To facilitate the sparse-reward training, we adapted standard MARL benchmarks, VMAS and SMACv2, into sparse-reward configurations as detailed below. This transformation creates challenging scenarios where intrinsic motivation is paramount for effective learning.

**VMAS Benchmarks.** For VMAS tasks that originally featured dense rewards (e.g., based on distance-to-target or control efforts), we induced sparsity by identifying a clear success condition for each task and restructuring the reward mechanism accordingly. Specifically, original dense or intermediate reward components were removed or nullified, except for a significant positive reward granted *only* upon the achievement of the predefined success condition (e.g., reaching a target zone, achieving a formation). Episode termination conditions remained largely unchanged, but rewards became predominantly contingent on successful terminal states. For instance, in a Balance* task, agents receive a large positive reward only upon moving the object to the goal region, with zero or minimal rewards during transit. This approach ensures that learning signals are tied to meaningful task completion rather than continuous incremental progress.

**SMACv2 Benchmarks.** Similarly, for SMACv2 scenarios, while many already possess somewhat sparse rewards, we further amplified sparsity to stringently test exploration. The primary reward signal was strictly tied to the ultimate battle outcome: a large positive reward for winning, a negative or zero reward for losing, and a neutral or slightly negative reward for a draw. All intermediate battlefield rewards, such as bonuses for damaging enemy units or penalties for sustaining damage, were removed. This concentrates the learning signal at the conclusion of an engagement, compelling agents to learn long-term coordination and strategy based purely on the sparse win/loss feedback, especially in scenarios where incremental damage yields no immediate reward.

These modifications ensure that agents in both benchmarks must explore extensively to discover successful action sequences, as intermediate feedback is largely absent, thus providing a robust testbed for curiosity-driven mechanisms like CERMIC.

## F    Supplementary Experimental Results

**Performance Curve Comparison.** We present further comparative experiments showcasing the performance of CERMIC against other methods in Fig. 4 below. The results demonstrate that CERMIC consistently surpasses prior SoTA models and other curiosity-driven approaches, achieving superior task performance. Furthermore, it is observable that our method exhibits more stable learning curves, characterized by narrower error bands. We attribute this enhanced stability and performance to CERMIC's ability to effectively filter noisy signals while concurrently encouraging robust exploration.

**Robustness to Noise.** To further validate the noise-filtering capabilities of CERMIC, we conducted comparative experiments in environments augmented with "random box" noise, as depicted in Fig. 5. The results indicate that, in these noisy conditions, CERMIC maintains more stable learning curves

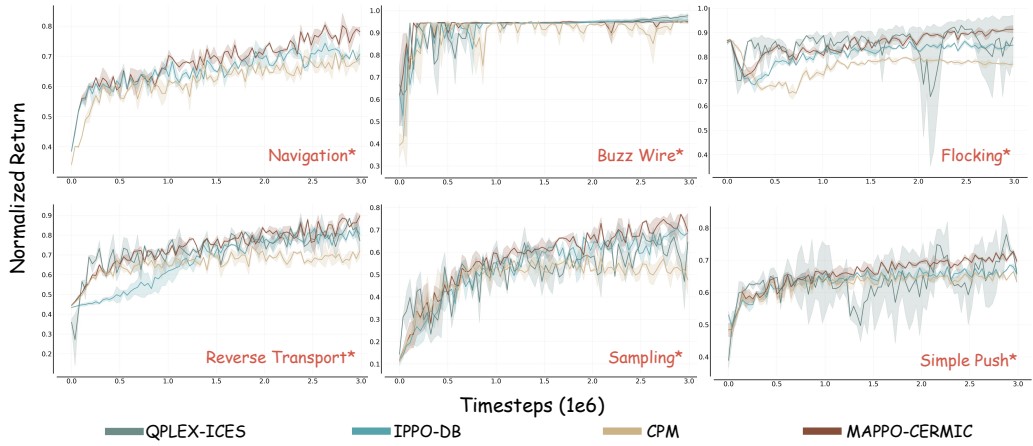

Figure 4: Comparative performance on VMAS.

and narrower error bands compared to other algorithms. Notably, this advantage in stability is more pronounced than in the noise-free task counterparts (Fig. 4). Furthermore, we observe that the error bands in the `Simple Adversary` task are smaller than those in the `Ball Trajectory` task. This difference can be attributed to the richer multi-agent contextual information available in the former, which includes signals from both adversaries and teammates, whereas the latter primarily involves information from a single partner. Consequently, the increased contextual feedback translates to a stronger noise-filtering effect, as reflected in the learning curves. This underscores CERMIC's ability to leverage multi-agent information to enhance task understanding and mitigate the impact of environmental stochasticity.

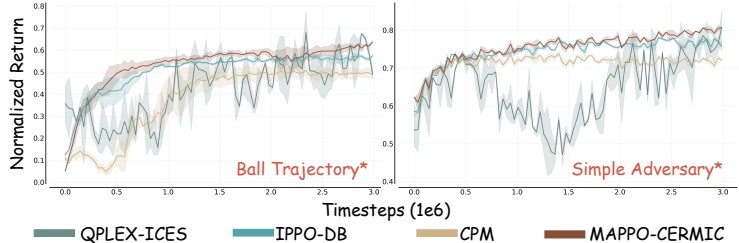

Figure 5: Performance comparison under environmental noise on VMAS task.

**Role Diversity and Task Adaptation.** While CERMIC consistently improves performance, we observe a more substantial performance gain over traditional baselines in tasks involving diverse agent roles (e.g., both cooperation and competition, as in `Simple Adversary`*) compared to purely cooperative/competitive settings (e.g., `Ball Trajectory`*); see Fig. 5. We attribute this to the challenge faced by traditional methods in interpreting heterogeneous agent behaviors under partial observability, where inferring intent and role from observations is inherently difficult. In contrast, CERMIC's task-adaptive weighting mechanism ($\gamma$) enables agents to better infer and adapt to others' intentions and plans, thereby enhancing performance in complex social interactions.

**Visualization of Intrinsic Reward.** To illustrate the temporal behavior of CERMIC's intrinsic reward, we visualized its values throughout a complete episode of the `Balance`* task, as shown in Fig. 6. A clear trend is the gradual decrease of this intrinsic reward over time, indicating a diminishing role of curiosity-driven exploration as the agent gains familiarity with the task environment. Furthermore, peaks in the intrinsic reward consistently coincide with an agent encountering novel situations, such as initial environmental entry or first-time interactions with other agents. This observation aligns with our design objective for the intrinsic reward, which is to incentivize exploration of unfamiliar states and interactions.

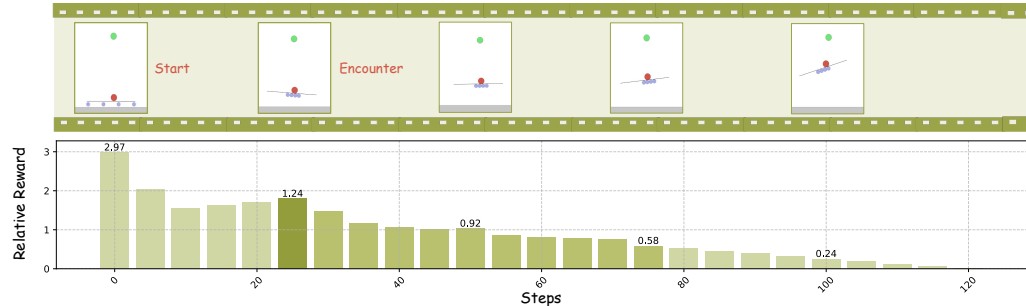

Figure 6: Visualization of CERMIC's intrinsic reward during one episode of the `Balance*` task.

## G  Alternative Memory Architectures

While graph-based memory offers structured relational reasoning, alternative architectures provide different trade-offs.Table 5 compares key architectural parameters with typical example values for these memory types.

**MLP-based Memory.** An MLP processes the agent's current (potentially encoded) observation to output a fixed-size memory vector. This is a stateless, feed-

**GRU-based Memory.** A Gated Recurrent Unit (GRU) maintains a hidden state that is updated at each timestep based on the current encoded observation and its previous state, capturing temporal dependencies.

Table 5: Architectural parameters for different memory modules.

| Parameter Type | Graph-based | MLP-based | GRU-based |
|---|---|---|---|
| **Core Encoding** | | | |
| Node Embedding Dim ($D_{node}$) | 256 | - | - |
| Edge Embedding Dim ($D_{edge}$) | 16 | - | - |
| Obs. Encoder Output Dim ($D_{obs\_enc}$) | 64 (for detection/encoding) | 64 | 64 |
| Memory/State Output Dim | 512 ($D_{GNN\_out}$) | 256 ($D_M$) | 256 ($D_H$) |
| **Network Structure** | | | |
| Obs. Encoder (CNN) Layers/Depth | 3 conv layers | 3 conv layers | 3 conv layers |
| GNN Layers ($L_{GNN}$) | 2 | - | - |
| GNN Hidden Dim ($D_{GNN\_hid}$) | 256 | - | - |
| GNN Heads ($N_{heads}$) | 4 | - | - |
| MLP Hidden Layers ($L_{MLP}$) | (Node/Edge Encoders: 2) | 2 (post obs-encoder) | - |
| MLP Hidden Units/Layer ($U_{MLP}$) | (Node/Edge Encoders: 128) | 128 | - |
| GRU Layers ($L_{GRU}$) | - | - | 2 |
| **Est. Total Parameters** | High | Low | Medium |

The Graph-based module is generally the most expressive due to its explicit relational structure but also tends to be the most parameter-heavy. MLP-based memory is the simplest, while GRU-based memory offers a balance for capturing temporal context. The choice depends on the specific requirements of the MARL task, including the complexity of inter-agent interactions and available computational resources.

## H  Implementation Details

For brevity, we previously use abbreviated task names in Table 1. Their full names are as follows: `Disper` = `Disperse`, `Naviga` = `Navigation`, `Sampli` = `Sampling`, `Passag` = `Passage`, `Transp` = `Transport`, `Balanc` = `Balance`, `GiveWa` = `GiveWay`, `Wheel` = `WheelGathering`, `Flocki` = `Flocking` (*: sparse-reward version), `StaHun` = `StagHuntRepeated`, `CleaUp` = `Cleanup`, `ChiGam` = `ChickenGameRepeated`, `PriDil` = `PrisonersDilemmaRepeated`, `pro5v5` = `Protoss5v5`, and `zer5v5` = `Zerg5v5`.

Table 6: Key training hyperparameters.

| Parameter Category | Value / Setting |
| --- | --- |
| *General Optimization* | |
| Discount Factor ($\gamma$) | 0.99 |
| Learning Rate (Adam) | 1e-4 |
| Adam Optimizer $\epsilon$ | 1e-6 |
| *Target Network Updates* | |
| Update Type | Soft (Polyak averaging) |
| Polyak Tau ($\tau$) | 0.005 |
| *Exploration Strategy* | |
| Initial Epsilon ($\epsilon_{\text{init}}$) | 0.8 |
| Final Epsilon ($\epsilon_{\text{end}}$) | 0.01 |
| *Training Duration* | |
| Max Frames | 3000000 |
| *On-Policy Collection & Training* | |
| Collected Frames per Batch | 36000 |
| Environments per Worker | 60 |
| Minibatch Iterations | 30 |
| Minibatch Size | 2400 |
| *Off-Policy Collection & Training* | |
| Optimizer Steps per Collection | 1800 |
| Train Batch Size | 1024 |
| Replay Buffer Size | 1500000 |

This paragraph outlines the core hyperparameters used for training our models and baselines, unless specified otherwise for particular tasks or algorithms. Specific parameters related to intrinsic rewards (e.g., exploitation weight factor $\alpha$) are detailed separately in the experimental setup as they vary across tasks and model configurations. Training/evaluation was conducted via BenchMARL suite on NVIDIA Quadro RTX 8000 GPUs.

