# OpenReview forum: "Wonder Wins Ways: Curiosity-Driven Exploration through Multi-Agent Contextual Calibration"
_NeurIPS.cc/2025/Conference — NeurIPS 2025 poster_

### Official Review · Reviewer_VKXv · 2025-07-02

**Clarity:** 2
**Significance:** 3
**Originality:** 3
**Rating:** 5
**Confidence:** 3

**Summary:**

CERMIC addresses exploration in decentralized, communication-less multi-agent reinforcement learning under sparse rewards. The authors identify the failure modes of standard novelty-driven intrinsic rewards, especially under environmental stochasticity and partial observability; they propose CERMIC: a plug-and-play module that uses a learned multi-agent intention model  to calibrate intrinsic curiosity signals. The method optimizes an Information-Bottleneck objective to maximize mutual information with future states while constraining self-information conditioned on past states and actions. Based on this objective and Bayesian surprise, CERMIC generates theoretical intrinsic rewards and integrates with standard MARL algorithms(MAPPO and QMIX). Empirically, CERMIC achieves state-of-the-art performance on 13 of 16 sparse-reward tasks across VMAS, MeltingPot, and SMACv2 benchmarks.

**Questions:**

1. Some recent works propose alternative mutual information measures (e.g. sliced mutual information [1,2]) that are more tractable and might work better in optimization and representation learning problems, have you considered incorporating these with your method? Could you provide a few comments on these and how your method (eg intrinsic reward) changes in the light of the new information measure?
2. How sensitive is CERMIC to the quality of the pre-trained agent detector? Could you report performance when using only on-policy learned detectors without ground-truth supervision?

[1] “Sliced Mutual Information: A Scalable Measure of Statistical Dependence” (Goldfeld & Greenewald, NeurIPS 2021)

[2] “On Slicing Optimality for Mutual Information” (Fayad & Ibrahim, NeurIPS 2023)

**Ethical Concerns:**

["NO or VERY MINOR ethics concerns only"]

**Limitations:**

yes

**Quality:**

3

**Strengths And Weaknesses:**

**Quality**

The paper is technically sound and presents an intriguing well-motivated theoretical framework based on the information bottleneck (IB) and robust chance constraints. They show how the intrinsic reward is connected to Bayesian surprise and aligned with UCB bonuses in linear MDPs. There has been an extensive empirical evaluation on diverse benchmark suites under standardized sparse-reward settings where wee see improvements over SoTA baselines. However, the reliance on learning an accurate intention model from scratch slows early training and may be impractical in very large agent populations. The pre-training procedure for the detector and graph memory uses privileged ground-truth labels, thus possibly limiting applicability to domains without such supervision.


**Clarity**

The paper is clearly structured, with intuitive diagrams, however the notation was dense that it might impede reader’s accessibility.

**Originality**

The combination of robust chance-constrained Information Bottleneck with multi-agent intention modeling is, as far as I know, original. The theoretical link between Bayesian surprise reward and UCB bonuses provides fresh insight.
However, Elements such as InfoNCE-based mutual information bounds and dynamic graphs have appeared in earlier works [eg arXiv:1807.03748]; thus the combination, while novel, builds on many established components.

**Significance**

This work tackles multi-agent exploration under sparse rewards which is a critical open problem; and as mentioned above: introducing socially contextualized intrinsic motivation represents a novel direction that will likely influence future MARL research.
That being said, the requirement for a pre-trained agent detector and graph memory (as described in Appendix A) may limit immediate impact in fully unknown environments. And the method’s complexity may hinder adoption compared to simpler curiosity baselines.

Overall, I found this work to be interesting and potentially impactful with a few limitations.

---

> ### Author Rebuttal · Authors · 2025-07-30
>
> We sincerely thank the reviewer for the thoughtful and encouraging feedback. We are grateful for the positive assessment of our work's technical soundness, originality, and extensive empirical evaluation. We address the specific weaknesses (W1, W2, W3) and questions (Q1, Q2) below.
>
> ---
> ### **W1: Reliance on Pre-training Procedure**
> We acknowledge this as an important practical consideration. However, We would like to clarify that the primary purpose of our pre-training experiments is to illustrate the significance of the intention modeling component and to highlight the broad, practical applicability of the CERMIC framework. We are motivated by the observation that the field of motion and behavior prediction is relatively mature, with many open-source models and APIs available. Our results demonstrate that CERMIC's performance can be further enhanced by leveraging these readily available, off-the-shelf tools, making it highly adaptable to real-world scenarios. Furthermore, we respectfully clarify that, unless explicitly stated otherwise, all primary results for CERMIC reported in the main text are from models trained from scratch, WITHOUT any pre-training.
>
> Fundamentally, our core contribution is the principled curiosity calibration mechanism that operates on top of an intention prediction; this mechanism is agnostic to the specific implementation of the intention predictor.
>
> ---
> ### **W2: Notation Accessibility**
> We thank you for pointing this out and will make several revisions to improve the paper's clarity and accessibility. To address this, we will make the following concrete revisions in the camera-ready version to improve readability and accessibility for a wider audience:
> 1. Add a comprehensive notation table in the Appendix to serve as a quick reference.
> 2. Expand on the background of chance constraint optimization in Section 4.2 to make the text more self-contained for a broader audience.
> 3. Perform a thorough proofreading of the entire manuscript to correct minor errors and refine phrasing for better readability.
>
> ---
> ### **W3: On the Manageable and Necessary Complexity of CERMIC**
> We will demonstrate the manageable nature of our method's complexity from both theoretical and empirical perspectives.
>
> ***Theoretically***, CERMIC is designed for efficiency. As illustrated in our workflow (Fig. 1), it first projects high-dimensional raw observations into a low-dimensional latent space. All subsequent, more complex computations are performed within this compact space, which inherently limits the computational burden. We further analyze the computational complexity. The primary overhead lies in the GNN used for intention modeling, while other components incur only constant-time costs (see Appendix Tables 4 and 5 for parameters). The GNN has a complexity of $O(N_o^2)$, where $N_o$ is the number of agents currently detected. However, due to partial observability, the number $N_o$ is bounded, and the overall computation remains manageable.
>
> ***Empirically***, to provide concrete evidence, we have conducted new profiling experiments measuring runtime and memory overhead. We compared CERMIC agents against SoTA method QPLEX-ICES on the VMAS benchmark as below. The results show that CERMIC remains a modest memorial cost (MAPPO-CERMIC: 10.4MB, 83% of QLPEX-ICES's weight under the same visual perception module) while achieving SoTA performance, underscoring CERMIC's effectiveness as an efficient, plug-and-play module.
> |Inference Time (per step: ms)|Balanc*|Disper*|Flocki*|
> |-|-|-|-|
> |QPLEX-ICES|2.6|2.1|3.7|
> |MAPPO-CERMIC|2.5|1.8|3.5|
>
> Furthermore, we argue that this manageable complexity is necessary, as our experiments reveal critical two failure modes in simpler curiosity baselines:
> 1. **Susceptibility to Social Distraction**: Our empirical results indicate that agents driven by simpler curiosity formulations (without explicit modeling of others) are persistently distracted by peers' actions in later training stages. From a single agent's perspective, a peer's complex behavior without inferred intent is treated as unlearnable noise, which a novelty-seeking algorithm is drawn to, thus hindering task progress. Our calibration mechanism transforms this noise into a learnable signal.
>
>     To further analyze this, we conducted additional experiments to examine how the contribution of each agent to the final performance as the number of agents increases. We selected the Balanc* scenario and used the task return weighted by the number of agents ($total$-$return$/$num$-$agent$) to represent the per-agent contribution. We find that in traditional methods, the per-agent contribution actually decreases as the number of agents increases, whereas NeuRO is able to mitigate this trend.
>
>     |Num. Agents|2|4|6|8|
>     |-|-|-|-|-|
>     |MAPPO-DB|12.3|13.7|13.6 | 12.1 |
>     |MAPPO-CERMIC|14.4|16.7 | 16.4 | 17.3 |
>
> 2. **Inability to Leverage Social Learning**: Beyond mitigating interference, we found that explicitly modeling peer intentions accelerates learning by enabling agents to understand implicit task rules and coordinate. This is validated by our pre-training ablation (Fig. 2), where agents with a proficient intention model converge significantly faster, mirroring how humans leverage team observation to learn and cooperate.
>
> ---
> ### **Q1: Alternative Mutual Information Measures**
> We sincerely thank you for this insightful question. The methods you've pointed to, Sliced Mutual Information (SMI) and particularly its optimized variant, Optimal SMI (ST*), represent an elegant approach to the challenges of high-dimensional mutual information (MI) estimation. We will focus our discussion on integrating the more advanced ST*
>
> The central idea of our current method is to learn a multi-agent contextual representation and use it to calibrate an exploration objective based on classic MI. This process is to learning an external auxiliary representation to better optimize a fixed, theoretically-optimal but computationally-intractable MI objective.
>
> The philosophy behind ST*, however, is fundamentally different. It posits that the uniform random projections of standard SMI are suboptimal, as they waste computational resources on uninformative "noisy" directions. The core contribution of ST* is therefore to learn an optimal distribution of projections, concentrating the information measure itself on the most critical subspaces. Therefore, integrating ST* into CERMIC would induce a fundamental shift in the curiosity calibration mechanism:
>
> **Current Framework**: `[Representation Learning Module] → Calibrates → [Fixed MI Objective]`
>
> **New Framework with ST***: `[Representation Learning Module] → Guides → [Adaptive ST* Objective]`
>
> In this new framework, CERMIC's multi-agent context module would no longer be calibrating a static information objective. Instead, it would be tasked with dynamically guiding the learning process of the "optimal slices" within ST*. For instance, the inferred multi-agent context could serve as a prior, helping ST* to more quickly discover projection directions that are not only critical for predicting future states, but also for understanding multi-agent interactions.
>
> As for the intrinsic reward, the integration of ST* would also lead to a more profound form of the reward signal. Our current intrinsic reward is derived from the Bayesian Surprise on CERMIC parameters. In the new framework, the primary learnable component within the reward would be the slicing policy network of ST*. Therefore, the intrinsic reward could be redefined on the parameters of this slicing policy network.
>
> In summary the integration would create a powerful synergistic effect: CERMIC provides the high-level social context, and ST*, under this guidance, efficiently identifies the most information-rich low-level subspaces. This combination of "high-level semantics guiding low-level representation learning" could be more robust and efficient than our current approach.
>
> ---
> ### **Q2: Sensitivity to the Pre-Trained Agent Detector**
> We wish to clarify that, UNLESS explicitly stated otherwise, all main results for CERMIC reported in the text use an on-policy learned detector without any pre-training to ensure a fair comparison.
> Our experiments (Fig. 2 in the main text and Fig. 8 in the Appendix) show that pre-training does lead to moderate performance improvement. However, it also introduces additional training overhead. This suggests that the on-policy learned graph and intention model are sufficient for achieving high performance after a reasonable amount of training.
> To provide additional support for this observation, we present numerical results. The table below compares the final performance of CERMIC with and without pre-training on several key tasks. The results are consistent with the conclusion above.
> ||Flocki*|Naviga|Passag*|CleanUp|ChiGam|
> |-|-|-|-|-|-|
> |CERMIC + pretrain|1.08 (±0.14)|1.46 (±0.07)|173 (±2.29)|83.8 (±0.60)|11.93 (±0.44)|
> |CERMIC|1.06 (±0.12)|1.43 (±0.06)|171 (±3.72)|78.1 (±0.82)|10.02 (±0.65)|

---

> > ### Comment · Reviewer_VKXv · 2025-08-06
> >
> > Thank you for the detailed responses, I have read the other reviews and rebuttals and still think the paper should be accepted and expect you to incorporate the discussions especially in W3 and Q1 as it will make the paper more complete.

---

> > > ### Author Response · Authors · 2025-08-07
> > >
> > > Thank you for your support and for taking the time to consider the other reviews and our rebuttals. We appreciate your suggestion and will make sure to incorporate the discussions in the final version to improve the completeness of the paper.

---

### Official Review · Reviewer_NuKT · 2025-07-02

**Clarity:** 1
**Significance:** 3
**Originality:** 3
**Rating:** 5
**Confidence:** 4

**Summary:**

This paper introduces CERMIC, a framework for intrinsically-motivated MARL in a decentralized, communication-less, sparse reward setting. CERMIC includes a GNN-based context learning mechanism that, intuitively, models agent intentions with an interesting information bottleneck for representation learning intended to deal with stochasticity in the environment. It presents both a theoretical analysis under linearity assumptions as well as experiments in several MARL environments, including both comparisons to MARL and intrinsically motivated MARL baselines as well as ablations.

**Questions:**

-Clarify why the loss ablation on Wheel is sufficient to demonstrate robustness to noise?

-Am I missing an eval environment, or were there 15? Do you get SotA on 13 or 12 environments?

-Could you explain how the pre-training process works? Does this make for an unfair comparison with baselines?

**Ethical Concerns:**

["NO or VERY MINOR ethics concerns only"]

**Final Justification:**

Provided in a separate comment.

**Limitations:**

yes

**Quality:**

2

**Strengths And Weaknesses:**

Strengths:

+Quality: the paper includes extensive experimentation on a number of benchmark environments, both in terms of baseline comparisons and ablations. The analyses/visualizations are very helpful.

+Significance: the experimental results demonstrate clear success on several benchmark environments, including challenging melting pot mixed motive environments.

+Originality: a number of the components to the architecture are, to my knowledge, new. I’m in particular not familiar with anything that uses this chance constraint component.

Weaknesses:

-Clarity: The paper is quite hard to read at times, leading to confusion that I think could be cleared up by editing. Some of these are small, e.g. l52 “given coverage level” is hard to understand in that surrounding text, l120 “updated via a momentum moving of o_t’s ones”, l144 “for conciseness, let…” when the term was introduced in an earlier equation, section 5.3 has two
”qualitative analysis” subsections, l155 “2e”, Eq 7 should I assume have “exploit” not “explore”. Others are more significant, e.g. the paper mentions 16 evaluation scenarios, whereas as far as I can tell, there are 15.

-Significance: Continuing with the headline results, the main text notes SotA performance on 13 out of 16 scenarios, but as I mentioned above, there appear to b 15 scenarios with 12 bolded (ours) results, and a baseline on zer5v5 ties your best on one but is not bolded. The rest appear not to hav errors like that, however, it’d be helpful to understand which differences are significant e.g. Disper on VMAS is very close. Results still look good but appear overstated.

-Significance/Quality: Ablations appear to be run on a small subset of environments, e.g. loss module ablations are run just on Wheel. I wouldn’t necessarily ask for more (you’ve run a lot of experiments) but it’d be helpful to understand why these choices make sense. E.g. the ablation on Wheel appears to be the primary supporting evidence that the explore/exploit loss configuration helps the model deal with noise, and a good deal of exposition was devoted to this property early on.

-Significance/Clarity: you use pre-training, but the particulars of how this is done are barely mentioned in the main text and not super clear in the supplementary, either. One concern is that this makes for an unfair comparison with baselines, as it relies on e.g. pre-training agent identifiers using ground truth signal.

---

> ### Author Rebuttal · Authors · 2025-07-30
>
> We sincerely thank you for the thorough and detailed feedback. We appreciate the positive acknowledgment of our work's originality and the extensive nature of our experiments. In response to your valid concerns, we have conducted new experiments and will make substantial revisions to the manuscript to address every point. We address the weaknesses (W1, W2, W3, W4) and questions (Q1, Q2, Q3) below.
>
> ---
> ### **W1: Clarity**
> We sincerely apologize for the lack of clarity and the numerous typographical errors; we have performed a thorough revision to address all identified issues and improve overall readability. Your meticulous reading has been invaluable for improving the quality of our manuscript.
>
> We have carefully corrected all specific errors you pointed out:
> - l155: "2e" → "we",
> - Eq 7: "explore" → "exploit",
> - Second “Qualitative Study” corrected to "Quantitative Analysis").
> - l52 "coverage level": We respectfully clarify that this is standard terminology in chance-constrained optimization, referring to the probability $1-\epsilon$ that the constraint must hold.
> - l120 "momentum moving": We have revised this to clarify it refers to a standard momentum update for the target encoder, where its weights are a moving average of the online encoder's weights.
>
> Furthermore, to fully enhance readability and better contextualize our work, we will also implement the following concrete changes in the revised manuscript:
>
> 1. **Figure Readability**: We will revise the color schemes in all figures using distinct and colorblind-friendly palettes to ensure all data series are easily distinguishable.
> 2. **Notation and Terminology**: To ease the reading process, we will add a comprehensive notation table in the Appendix to serve as a quick reference. At the same time, we will present the relationships between modules in a more intuitive manner to clarify the underlying logic.
> 3. **Background Context**: To make the text more self-contained for a broader audience, we will expand on the background of key prerequisite concepts, notably chance-constrained optimization, in the relevant sections.
>
> ---
> ### **W2 & Q2: Experimental Rigor**
> You are correct, and we sincerely apologize for the typos and the lack of precision in our initial results presentation. We evaluated on 15 total scenarios and our method achieved SoTA performance in 12 of them. We have corrected this in the revised version. To address your valid and crucial concerns about significance and potential overstatement, we now provide a more rigorous, statistically-grounded analysis.
>
> ***Firstly***, to directly address whether our improved scores are genuine or merely due to chance, we will report the 95% confidence interval for the mean performance of all methods as follows (due to space limitations, we only show some of the results). The results show that the confidence interval of our method lies entirely above the confidence interval of a baseline, it provides strong statistical evidence that our performance is genuinely higher.
>
> Also, by examining the size of the confidence intervals in the revised table, one can observe that CERMIC's intervals are consistently tighter than those of the baselines. This demonstrates that CERMIC not only performs better on average but also converges more reliably and robustly.
> |          | Flocki*        | Naviga        | Passag*       | CleanUp       | ChiGam        |
> |-|-|-|-|-|-|
> | MAPPO         | -0.36 (±0.11)  | 1.07 (±0.10)  | 157 (±5.62)   | 74.2 (±0.93)  | 8.43 (±0.84)  |
> | QMIX-SPECTRA  | 0.01 (±0.14)   | 1.13 (±0.07)  | 155 (±4.38)   | 70.3 (±0.84)  | 8.50 (±0.79)  |
> | QPLEX-ICES    | 0.64 (±0.15)   | 1.34 (±0.07)  | 162 (±4.20)   | 76.5 (±1.06)  | 9.07 (±0.80)  |
> | MAPPO-CERMIC  | 1.06 (±0.12)   | 1.43 (±0.06)  | 171 (±3.72)   | 78.1 (±0.82)  | 10.02 (±0.65) |
>
> ***Secondly***, while achieving such superior performance, CERMIC maintains a moderate memory overhead (MAPPO-CERMIC: 10.4MB, 83% of QLPEX-ICES's weight under the same visual perception module), indicating that CERMIC can be lightweightly integrated into existing algorithms to enhance agent performance. CERMIC's lightweight yet promising performance gains highlight its practical potential for seamless integration into existing pipelines. Your request for experimental rigor was highly insightful and has greatly improved the presentation and reliability of our results.
>
> ---
> ### **W3 & Q1: Ablation Study**
> We thank you for this valuable suggestion. To thoroughly address this point, we have conducted extensive new ablation studies across diverse scenarios (8 seeds with 95% confidence interval), which robustly validate our claims.
> |                      |Flocki*     |Naviga     |Passag*     |CleanUp     |ChiGam     |
> |-|-|-|-|-|-|
> |**Loss Ablations ($\alpha=0.2$, Graph-Memory)**|||||
> |w/o $L_{explore}$        |0.82 (±0.15)|1.41 (±0.04)|166 (±3.66) |75.4 (±1.05)|9.22 (±0.80)|
> |w/o $L_{exploit}$       |0.72 (±0.48)|1.36 (±0.10)|164 (±4.41) |71.5 (±1.59)|8.47 (±1.18)|
> |MAPPO-CERMIC         |1.06 (±0.12)|1.43 (±0.06)|171 (±3.72) |78.1 (±0.82)|10.02 (±0.65)|
> |**Hyperparameter $\alpha$ (Graph-Memory)**|||||
> |1.0                  |0.80 (±0.07)|1.40 (±0.02)|163 (±1.56) |71.2 (±0.77)|9.04 (±0.46)|
> |0.5                  |0.88 (±0.10)|1.42 (±0.06)|169 (±3.78) |74.6 (±0.68)|9.50 (±0.67)|
> |0.2                  |1.06 (±0.12)|1.43 (±0.06)|171 (±3.72) |78.1 (±0.82)|10.02 (±0.65)|
> |**Intention Memory Type ($\alpha=0.2$)**|||||
> |GRU                  |0.78 (±0.27)|1.39 (±0.04)|161 (±3.15) |75.2 (±1.03)|9.19 (±0.97)|
> |Graph                |1.06 (±0.12)|1.43 (±0.06)|171 (±3.72) |78.1 (±0.82)|10.02 (±0.65)|
>
> The new results, which we will update in Figure 4 and Table 2 of the revised manuscript, consistently demonstrate that:
> 1. The results confirm that each loss component is essential to CERMIC's performance. This demonstrates that the exploit loss helps the agent better leverage learned knowledge and filter out noise, while the explore loss improves the agent’s overall performance in sparse-reward environments.
> 2. Employing more sophisticated memory modules yields better performance in complex scenarios.
>
> These consistent results across a diverse set of environments provide strong and convincing evidence for the necessity of each component in CERMIC's design.
>
> ---
> ### **W4 & Q3: Pre-training Procedure**
> The pre-training of our intention modeling module is conducted as follows. We deploy $N$ agents in the environment and allow them to move. At each timestep, the intention module for a given agent is tasked with producing two outputs for the other $N-1$ agents: (1) a judgment on whether each peer is within its observation field, and (2) an estimation of the state for any detected peers. These outputs—the presence judgment and the state estimation—are then supervised using the ground-truth information. The network is updated by computing losses via Eq. (14) and Eq. (15) in Appendix A, which correspond to a detection loss and a state prediction loss, respectively.
>
> We acknowledge your valid concern that introducing pre-training could lead to an unfair comparison with baselines. To ensure fairness, we want to clarify that, unless explicitly stated otherwise, all primary results for CERMIC reported in the main text (e.g., Table 1) are from models trained from scratch, WITHOUT any pre-training. The pre-trained variant is only used for a specific ablation study to analyze the impact of a proficient intention model. However, results show that the CERMIC agent without the pretraining module still achieves SoTA performance in 12 out of 15 scenarios. We provide additional experiments on the role of the pretraining module in Appendix Figure 7.
>
> The primary purpose of conducting experiments with a pre-trained module is to illustrate the significance of the intention modeling component and to highlight the broad applicability and future potential of CERMIC. The field of motion and behavior prediction is relatively mature, with many lightweight open-source models and APIs available. Our results demonstrate that CERMIC's performance can be further enhanced by leveraging these readily available, off-the-shelf tools. This modularity makes our framework highly adaptable to real-world scenarios where such auxiliary predictive models can be easily integrated to bootstrap performance.
>
> ---
>
> Once again, we express our sincere gratitude for your detailed and constructive review. We hope that our detailed responses and the new experimental results have fully addressed your concerns and have clarified the contributions and significance of our work. We look forward to your further feedback.

---

### Official Review · Reviewer_iVNP · 2025-07-03

**Clarity:** 3
**Significance:** 3
**Originality:** 3
**Rating:** 5
**Confidence:** 3

**Summary:**

In this work, the authors propose CERAMIC, a method for multiagent exploration which robustly filters noisy surprise signals and guides exploration by dynamically calibrating intrinsic curiosity with inferred multi-agent contextual awareness. CERAMIC also builds off of results from the single agent exploration community to adapt an approach for computing information gain from state transitions as a curiosity reward signal. CERAMIC achieves state of the art results on the benchmark suits VMAS, Meltingpot, and SMACv2 with particularly high performance in sparse reward settings.

**Questions:**

I understand the point about the clustering behavior around trajectory endpoints in the qualitative evaluation of the agent dynamics, but I do not understand the other points of significance in the qualitative analysis. Could you explain further what is significant about the agent state dynamics?

At first, I thought that the method was too similar to the intrinsic curiosity formulation proposed for the single agent case. I found that there were a lot of additional modeling considerations that you have to take into account. Could you explain why you would or would not necessarily expect to see the same relative performance and behavior in the multiagent setting if you use intrinsic rewards built and designed for single agents?

**Ethical Concerns:**

["NO or VERY MINOR ethics concerns only"]

**Final Justification:**

I thank the authors very much for their detailed response. I maintain my score and agree with the other reviewers on acceptance for all the points previously discussed.

In particular, I think that the authors did an excellent job in the rebuttal of describing the specific contributions significant to multiagent system dynamics that were unclear. I highly encourage in the final revision of this paper to include these descriptions as well as improvements to the figures for clarity.

**Limitations:**

Yes

**Quality:**

4

**Strengths And Weaknesses:**

The paper is well written and clear. The related work is thorough but some of the references could be introduced earlier. Specifically, the references to the very similar curiosity objective for the single agent case could have been discussed and contextualized in the related work. Two additional references to prior work with very similar intrinsic reward design in single agents:

Ruo Yu Tao, et al. Novelty Search in Representational Space for Sample Efficient Exploration. NeurIPS, 2020.
Bernadette Bucher, et al. Perception-driven Curiosity with Bayesian Surprise. RSS Workshop on Combining Learning and Reasoning Towards Human-Level Robot Intelligence, 2019.


The system has theoretical guarantees about exploration performance which the authors highlight. The intention graph is an interesting and novel methodological approach to modeling agent intention in the multiagent settings. The experiments in Table 1 seem to validate the claims of the paper as well as the model ablations. The training dynamics result was interesting and educational about the method.

I did not fully understand the point about the qualitative results for agent dynamics, and I think a clearer explanation is needed about what is desirable in multiagent system dynamics and how the results in the paper show this behavior. The line plots are very hard to read because the chosen colors are two close together. For example, in Figure 4, several very similar shades of blue are used.

The authors give full details for reproducing the method including code and all relative hyperparameters. Substantial additional experimentation is included in the appendix as well as in depth theoretical explanations of the system and its behavior. Overall, this paper is novel and interesting, technically deep, and experimentally thorough. It was also well written and educational to read.

---

> ### Author Rebuttal · Authors · 2025-07-30
>
> We sincerely thank you for the insightful and encouraging feedback. We are pleased that you found our work novel and interesting, technically deep, and experimentally thorough. We address the specific points on clarity (W1) and the questions (Q1, Q2) below.
>
> ---
> ### **W1: Clarity**
> We appreciate you pointing out this point for improvement. To enhance readability and better contextualize our work, and will implement the following concrete changes in the revised manuscript:
> 1. **Figure Readability**: We will revise the color schemes in Figures 3, 4, and 6 using distinct and colorblind-friendly palettes to ensure all data series are easily distinguishable.
> 2. **Related Work Context**: We thank you for pointing us to the two highly relevant papers (Tao et al., 2020; Bucher et al., 2019). We agree they are important and will integrate them into our Related Work section to better contextualize our work and highlight the transition from single-agent to multi-agent curiosity.
> 3. **Notation and Background**: To further improve clarity, we will add a comprehensive notation table in the Appendix and briefly expand on the background of chance constraint optimization in Section 4.2 to make the manuscript more self-contained.
>
> ---
> ### **Q1: Elaborating on the Significance of State Dynamics (Figure 3)**
> The qualitative analysis in Figure 3 aims to demonstrate how CERMIC fosters a more sophisticated, human-like curiosity, which manifests in four key phenomena. The first phenomenon illustrates curiosity towards peers (social curiosity), while the subsequent three illustrate curiosity towards the environment (environmental curiosity).
>
> **Observed in the left panels of Figure 3 (Social Curiosity)**
>
> ***Phenomenon 1. Intentional Trajectory Crossing***: Unlike conventional MARL agents (e.g., MAPPO agents in the figure) whose state trajectories tend to run in parallel (indicating mutual ignorance), CERMIC agents exhibit converging-then-diverging trajectory paths (forming an 'X' shape). This signifies that agents will first intentionally approach a peer to observe, and then diverge to execute their own plans. Crucially, the trajectories do not fully merge into a Y-shape, which would imply mere imitation. This behavior demonstrates an advanced social curiosity, where agents strategically leverage peer observation to inform their own decision-making.
>
> **Observed in the right panels of Figure 3 (Environmental Curiosity)**
>
> ***Phenomenon 2. Endpoint Clustering (as you noted)***: The clustering of latent states $x_t$ at the endpoints of state trajectories indicates that the curiosity module is most active during significant state transitions, which represent novel situations. This aligns with the human tendency to be most curious when facing new and unfamiliar circumstances.
>
> ***Phenomenon 3. Latent State Dispersion***: We observe that for any given state $s_t$ (the blue point), the corresponding latent states $x_t $ (green points) do not collapse onto $s_t$ but rather form a dispersed cloud around it. This signifies that the agent's curiosity is directed not at its current, known state, but at exploring the adjacent, unvisited states in its local state manifold. This is a targeted and sample-efficient form of exploration, akin to a human's curiosity about what lies "just around the corner".
>
> ***Phenomenon 4. Intrinsic Reward Dynamics***: The right panel shows the agent's intrinsic reward value. For context, the two blue spheres in the "Global Task State" are the agents, and the red sphere is the object they need to transport. We observe a marked increase in intrinsic reward when the two agents approach each other for cooperative action. This empirically validates that our module successfully identifies and incentivizes agent’s attention towards their peers during critical interactive moments, consistent with our design philosophy.
>
> ---
> ### **Q2: Distinguishing CERMIC from Single-Agent Curiosity**
> This is a crucial question that gets to the heart of our contribution. We believe directly applying single-agent intrinsic rewards to MARL settings is suboptimal due to their inability to handle socially-induced stochasticity and leverage social learning opportunities. We explain this based on two key observations from our experiments and analysis:
>
> 1. **Susceptibility to Social Distraction**: We observe that agents driven by single-agent curiosity formulations (e.g., MAPPO-DB), which do not explicitly model others, are persistently distracted by their peers' actions, even in later training stages. From a single agent's perspective, a peer's complex behavior, without inferred intent, can be viewed as high-entropy, socially-induced stochasticity. A novelty-based algorithm will thus compel the agent to pay undue attention to this "noise", hindering task progress (compared to CERMIC, MAPPO-DB exhibits an almost 10% performance drop across nearly all tasks). Our contextual calibration mechanism is specifically designed to address this issue by transforming this perceived noise into a learnable signal, allowing agents to distinguish between meaningless randomness and intentional peer behavior.
>
>     To further analyze this, we conducted additional experiments to examine how the contribution of each agent to the final task performance changes as the number of agents increases. We selected the **Balance*** task scenario and used the task return weighted by the number of agents ($total$-$return$/$num$-$agent$) to represent the per-agent contribution. We find that in traditional methods, the per-agent contribution actually decreases as the number of agents increases, whereas NeuRO is able to mitigate this trend.
>
>     | Num. Agents | 2    | 4    | 6    | 8    |
>     |-|-|-|-|-|
>     | MAPPO-DB   | 12.3 | 13.7 | 13.6 | 12.1 |
>     | MAPPO-CERMIC | 14.4 | 16.7 | 16.4 | 17.3 |
>
> 2. **Inability to Leverage Social Learning**: Beyond mitigating interference, explicitly modeling others’ intentions accelerates learning by enabling agents to infer task rules from their peer behaviors, thus enhancing task performances. This is further validated by our pre-training experiment (Fig. 2), where CERMIC agents initialized with a pre-trained intention model exhibit significantly faster convergence. These results, as highlighted by your insightful questions, underscore the importance of a robust mechanism for peer understanding in accelerating adaptation to task rules This mirrors human learning, where observing teammates is crucial for understanding a task and coordinating effectively. Single-agent methods inherently lack this capability.

---

### Official Review · Reviewer_ckU6 · 2025-07-03

**Clarity:** 4
**Significance:** 3
**Originality:** 3
**Rating:** 4
**Confidence:** 4

**Summary:**

This paper investigates the exploration-exploitation problem in MARL from an information-theoretic perspective. Specifically, the authors design CERMIC, a plug-and-play module that helps multi-agent systems learn curiosity-driven signals, and integrates with existing algorithms in the form of intrinsic rewards. Experiments on several benchmarks demonstrate that the proposed method achieves SOTA performance in sparse reward scenarios.

**Questions:**

1.Besides intrinsic rewards, what other approaches (of utilizing CERMIC) does the author believe can improve the training effectiveness of MARL? I would like the author to provide an in-depth analysis on this question.

2.How should the ∪ symbol in Equation 12 be interpreted?

**Ethical Concerns:**

["NO or VERY MINOR ethics concerns only"]

**Final Justification:**

Most of my concerns addressed.

**Limitations:**

Yes.

**Paper Formatting Concerns:**

No related concerns at this time.

**Quality:**

3

**Strengths And Weaknesses:**

**Strengths**:

(Major) The sparse reward problem is a commonly challenging issue in RL, while the exploration-exploitation problem represents a more fundamental challenge in RL. The proposed novelty-based exploration and Multi-Agent Contextualized Exploitation effectively address the corresponding problems in the MARL domain.

(Minor) This paper is well-written with clear logic and reasonable structure, making it smooth to read.

(Minor) The experimental evaluation is comprehensive, comparing against SOTA methods and curiosity-based approaches in the field, and achieving new SOTA performance. Additionally, the well-designed visualization experiments enhance the overall clarity of the paper.

---

**Weaknesses**:

(Minor) The individual ablation study (Section 5.4) is conducted only in one scenario of one benchmark, which lacks convincing evidence. I suggest the authors conduct experiments across all scenarios within one benchmark or select representative scenarios from all benchmarks.

(Minor) The proposed method introduces additional computational overhead and complexity. How do the authors analyze this aspect?

---

> ### Author Rebuttal · Authors · 2025-07-30
>
> We sincerely appreciate your thoughtful and detailed feedback. We are encouraged by your recognition of our work’s novelty, clarity, and empirical strength. Below, we respectfully address the identified weaknesses (W1, W2) and respond to the insightful questions (Q1, Q2).
>
> ---
> ### **W1: Ablation Study**
> We thank you for this valuable suggestion. To thoroughly address this point, we have conducted extensive new ablation studies across diverse scenarios (8 seeds, 95% confidence interval), which robustly validate our claims.
> |                      |Flocki*     |Naviga     |Passag*     |CleanUp     |ChiGam     |
> |-|-|-|-|-|-|
> |**Loss Ablations ($\alpha=0.2$, Graph-Memory)**|||||
> |w/o $L_{explore}$        |0.82 (±0.15)|1.41 (±0.04)|166 (±3.66) |75.4 (±1.05)|9.22 (±0.80)|
> |w/o $L_{exploit}$       |0.72 (±0.48)|1.36 (±0.10)|164 (±4.41) |71.5 (±1.59)|8.47 (±1.18)|
> |MAPPO-CERMIC         |1.06 (±0.12)|1.43 (±0.06)|171 (±3.72) |78.1 (±0.82)|10.02 (±0.65)|
> |**Hyperparameter $\alpha$ (Graph-Memory)**|||||
> |1.0                  |0.80 (±0.07)|1.40 (±0.02)|163 (±1.56) |71.2 (±0.77)|9.04 (±0.46)|
> |0.5                  |0.88 (±0.10)|1.42 (±0.06)|169 (±3.78) |74.6 (±0.68)|9.50 (±0.67)|
> |0.2                  |1.06 (±0.12)|1.43 (±0.06)|171 (±3.72) |78.1 (±0.82)|10.02 (±0.65)|
> |**Intention Memory Type ($\alpha=0.2$)**|||||
> |GRU                  |0.78 (±0.27)|1.39 (±0.04)|161 (±3.15) |75.2 (±1.03)|9.19 (±0.97)|
> |Graph                |1.06 (±0.12)|1.43 (±0.06)|171 (±3.72) |78.1 (±0.82)|10.02 (±0.65)|
>
>
> The new results, which we will update in Figure 4 and Table 2 of the revised manuscript, consistently demonstrate that:
>
> 1. These results confirm that each loss component is essential to CERMIC's performance. This demonstrates that the exploit loss helps the agent better leverage learned knowledge and filter out noise, while the explore loss improves the agent’s overall performance in sparse-reward environments.
>
> 2. Employing more sophisticated memory modules yields better performance in complex scenarios.
>
> These consistent results across a diverse set of environments provide strong and convincing evidence for the necessity of each component in CERMIC's design.
>
> ---
> ### **W2: Computational Complexity**
> We appreciate your attention to computational cost. We have carefully considered this point both from theoretical and empirical perspectives.
>
> ***Theoretically***, CERMIC is designed with computational efficiency in mind. As illustrated in our workflow (Fig. 1), the computationally intensive raw observations are first projected into a low-dimensional latent space. All subsequent CERMIC computations, including the multi-agent inference and chance constraint optimization, are performed within this compact latent space, inherently limiting the computational burden. We further analyze the computational complexity. The primary overhead lies in the GNN used for intention modeling, while other components incur only constant-time costs (see Appendix Tables 4 and 5 for parameters). The GNN has a complexity of $O(N_o^2)$, where $N_o$ is the number of agents currently detected. However, due to partial observability, the number $N_o$ is bounded, and the overall computation remains manageable.
>
> ***Empirically***, to further quantify this, we have conducted new profiling experiments measuring the runtime (inference time) and memory overhead. We compared CERMIC agents against SoTA method QPLEX-ICES on the VMAS benchmark as below. The results show that CERMIC remains a modest memorial cost (MAPPO-CERMIC: 10.4MB, 83% of QLPEX-ICES's weight under the same visual perception module) while achieving SoTA performance, underscoring CERMIC's effectiveness as an efficient module. CERMIC's lightweight yet promising performance gains highlight its plug-and-play capability.
> | Inference Time (per step: ms) | Balanc* | Disper* | Flocki* |
> |-|-|-|-|
> | QPLEX-ICES                    | 2.6     | 2.1     | 3.7     |
> | MAPPO-CERMIC                 | 2.5     | 1.8     | 3.5     |
>
> ---
> ### **Q1: Training Effectiveness**
> We thank you for raising this insightful and forward-looking question. In addition to intrinsic rewards, we believe the core elements of CERMIC—specifically the multi-agent contextual feature $f_n^o$ and the adaptive factor $\gamma$—can be leveraged to address other key challenges in MARL training. We envision two promising directions:
>
> 1. **Hierarchical Reinforcement Learning (HRL)**: In MARL, agents often need to adopt specific roles (e.g., cooperative or adversarial) based on the macroscopic tactical situation. The contextual feature $f_n^o$ generated by CERMIC is key to assessing this situation as it encapsulates inferred peer intentions. This would allow for a principled design of a two-level hierarchy where: (i) a high-level policy $\pi_{high}$ takes $f_n^o$ and the agent's state $s_t$ as input to select a subgoal or role $g_t$; (ii) a low-level policy $\pi_{low}$ then executes primitive actions based on $g_t$ and $s_t$ to accomplish the current subgoal. This hierarchical design leveraging multi-agent intention modeling would enable more structured and long-horizon strategic reasoning.
> 2. **Reward Credit Assignment**: For reward credit assignment, we propose a novel usage of $f_n^o$ to estimate how an agent is perceived by others. This facilitates fairer credit attribution and quicker role stabilization in cooperative scenarios. For instance, in a soccer game, how does an agent identify the most helpful teammate leading to a goal? The feature $f_n^o$ represents the current agent's judgment of others' intentions. It can help us understand what role other agents expect our agent to play. Based on this, in a cooperative game, if an agent is consistently perceived by others as a valuable collaborator, we can assign it a higher share of the team reward. This approach provides a novel mechanism for agents to quickly identify and settle into positions that maximally contribute to the team's success.
>
> In both scenarios, to handle the initial inaccuracy of the $f_n^o$ model, we can leverage the task-adaptive factor $γ$ to temper reliance on $f_n^o$ until the intention predicting module “matures”, ensuring stability during the early stages of training.
>
> ---
> ### **Q2: Symbol Interpretation**
> We thank you for pointing out the ambiguity of this notation. The symbol is intended to signify the addition of a new data point to the dataset of past experiences. This $\cup$ symbol denotes the action of incorporating the current transition $(s_t, a_t, s_{t+1})$ into the dataset $D_m$, which contains corresponding transitions from the past m episodes. To better illustrate this, we explain the meaning of the formula in which it appears: the purpose of the intrinsic reward in Eq. 12 is to measure the magnitude of change in the CERMIC module's parameter posterior after being updated with this new data. A larger reward implies a greater impact on the parameter distribution, indicating that the new data is novel and has not been encountered in the historical data from the past m episodes. This design sensitizes the CERMIC module to new situations, aligning with the fundamental goal of curiosity-driven exploration. To eliminate this ambiguity in the revised manuscript, we will first define the updated dataset $D_m^{\prime} = D_m \cup (s_t, a_t, s_{t+1})$, and then expressing the formulation of the intrinsic reward.

---

> > ### Comment · Reviewer_ckU6 · 2025-08-06
> >
> > Thank you for the authors' response.
> >
> > 1. The additional ablation experiments are good.
> >
> > 2. I accept the theoretical analysis, but could the empirical experiments add comparisons with more common algorithms, such as MAPPO, QPLEX, and QMIX? Additionally, could you compare the training time (besides inference time)?
> >
> > I will maintain my score and support the paper's acceptance.

---

> > > ### Author Response · Authors · 2025-08-07
> > >
> > > We thank you for your time and constructive feedback, which helped improve the quality of our work. We appreciate your suggestion regarding the training cost and more baseline algorithms. We will include these results in the revised version (due to limited space, we are currently able to include only a subset of the results).
> > > | Method        | Balance* - Training (s/iter) | Balance* - Inference (ms/step) | Flocking* - Training (s/iter) | Flocking* - Inference (ms/step) |
> > > |---------------|------------------------------|----------------------------------|-------------------------------|----------------------------------|
> > > | MAPPO         | 18.03                        | 2.2                              | 25.69                         | 3.3                              |
> > > | QMIX          | 24.35                        | 2.4                              | 33.34                         | 3.5                              |
> > > | MADDPG        | 26.98                        | 2.8                              | 34.76                         | 3.5                              |
> > > | MASAC         | 35.24                        | 3.0                              | 43.42                         | 3.9                              |
> > > | QPLEX-ICES    | 42.18                        | 2.6                              | 51.28                         | 3.7                              |
> > > | MAPPO-CERMIC  | 30.62                        | 2.5                              | 35.60                         | 3.5                              |
> > >
> > >
> > > Experiments show that while our method incurs higher training time due to the graph memory module, it remains manageable compared to previous SoTA methods. Furthermore, as noted in our theoretical discussion, the inference phase primarily involves low-dimensional computations, resulting in inference time comparable to that of traditional baselines. The results demonstrate the overall efficiency of our approach.

---

### Decision · Program_Chairs · 2025-09-17

**Decision:**

Accept (poster)

**Comment:**

This paper introduces CERMIC, a principled framework for curiosity-driven exploration in multi-agent reinforcement learning that calibrates intrinsic rewards using inferred contextual awareness of peers. The reviewers found the central idea both novel and timely, addressing the longstanding challenge of robust exploration under sparse rewards and partial observability. The method is theoretically motivated via the Information Bottleneck principle, and the authors provide clear algorithmic design along with ablations that highlight the importance of each component. Empirical results across VMAS, MeltingPot, and SMACv2 benchmarks show consistent and often state-of-the-art improvements over strong MARL baselines and prior curiosity-based methods, validating both effectiveness and generality. While limitations remain regarding scalability of intention modeling and reliance on local observations, the authors acknowledge these and suggest promising directions for future work. Overall, the paper makes a significant and well-substantiated contribution, and I recommend acceptance.